# Executioner caspases restrict mitochondrial RNA-driven Type I IFN induction during chemotherapy-induced apoptosis

Shane T. Killarney [1], Rachel Washart [1], Ryan S. Soderquist[1], Jacob P. Hoj [1], Jamie Lebhar[1], Kevin H. Lin[1,2] & Kris C. Wood [1] ✉

During apoptosis, mitochondrial outer membrane permeabilization (MOMP) enables certain mitochondrial matrix macromolecules to escape into the cytosol. However, the fate of mitochondrial RNA (mtRNA) during apoptosis is unknown. Here, we demonstrate that MOMP results in the cytoplasmic release of mtRNA and that executioner caspases-3 and -7 (casp3/7) prevent cytoplasmic mtRNA from triggering inflammatory signaling. In the setting of genetic or pharmacological casp3/7 inhibition, apoptotic insults result in mtRNA activation of the MDA5/MAVS/IRF3 pathway to drive Type I interferon (IFN) signaling. This pathway is sufficient to activate tumor-intrinsic Type I IFN signaling in immunologically cold cancer models that lack an intact cGAS/STING signaling pathway, promote CD8+ T-cell-dependent anti-tumor immunity, and overcome anti-PD1 refractoriness in vivo. Thus, a key function of casp3/7 is to inhibit inflammation caused by the cytoplasmic release of mtRNA, and pharmacological modulation of this pathway increases the immunogenicity of chemotherapy-induced apoptosis.

Intrinsic apoptosis is an evolutionarily conserved programmed cell death pathway employed by organisms to free themselves of deleterious cells[1]. MOMP is the governing step of intrinsic apoptosis and occurs when cytotoxic stress events overwhelm pro-survival BCL2 members, leading to the activation of BCL2 associated X, apoptosis regulator (BAX) and BCL2 antagonist/killer 1 (BAK)[2]. BAX and BAK form porous oligomers on the outer mitochondrial membrane, releasing inner membrane proteins into the cytosol, including cytochrome c[3]. Cytochrome c enters the cytoplasm following MOMP and forms the apoptosome complex with pro-caspase-9 and apoptotic protease activating factor-1 (Apaf-1). Auto-activation of caspase-9 occurs within the apoptosome and initiates the downstream cleavage of executioner casp3/7. Following the release of cytochrome c, BAX and BAK create macropores large enough to facilitate the herniation of the inner mitochondrial membrane into the cytoplasm[4,5]. The herniated inner membrane breaks down to create a communication channel between the mitochondrial matrix and cytosol capable of transmitting macromolecules, including nucleic acid species, between these cellular compartments. Although mtRNA is known to be immunogenic, in part owing to the fact that bidirectional transcription of the mitochondrial genome results in long, double-stranded RNA (dsRNA) species within the mitochondrial matrix[6–9], it is thought to be turned over rapidly by the degradosome components mitochondrial RNA helicase SUV3 and polynucleotide phosphorylase PNPase to avoid unwanted inflammation[8]. Thus, its localization and potential to activate cytosolic nucleic acid sensing pathways following chemotherapy-induced MOMP is unclear.

Type I IFNs are critical regulators of antiviral immunity and cancer immunosurveillance[10–12]. Type I IFNs, including IFN-α and IFN-β, are activated following the detection of damage-associated molecular patterns (DAMPs) by cytosolic or endosomal nucleic acid pattern recognition receptors (PRRs). IFN-α and IFN-β are exported into the extracellular space, where they bind to the IFNα receptor (IFNAR) in an autocrine and paracrine manner. IFNAR activation leads to the

---

[1]Department of Pharmacology and Cancer Biology, Duke University, Durham, NC, USA. [2]Present address: Department of Medicine, Brigham and Women's Hospital and Harvard Medical School, Boston, MA, USA. ✉e-mail: kris.wood@duke.edu

transcription of hundreds of interferon-stimulated genes (ISGs), which control an antiviral state. In cancer, Type I IFNs promote multiple steps of the cancer-immunity cycle, including the maturation of dendritic cells (DC), the cross-priming of tumor-associated antigens (TAAs) between DCs and CD8[+] T-cells, and the recruitment of cytotoxic T lymphocytes (CTLs) to the site of the tumor[13–15]. For these reasons, the generation of Type I IFNs in the tumor microenvironment (TME) is an attractive immunotherapeutic strategy in cancer[16–21]. However, efforts to pharmacologically activate Type I IFN production have suffered from important practical limitations. First, these efforts have largely focused on therapeutically activating Type I IFN signaling by engaging the cyclic GMP-AMP synthase (cGAS)-stimulator of interferon genes (STING) cytosolic dsDNA sensing pathway, either directly or through its upstream regulators[22]. Unfortunately, malignant cells are routinely selected for impaired cGAS/STING pathway activity during tumorigenesis, presumably to avoid immune detection[23–27]. For example, protein-level analyses of tumor specimens suggest that ~60% and ~50% of metastatic melanoma[23] and stage III colorectal cancers[24], respectively, lose cGAS/STING signaling competence through the repression of cGAS expression, STING expression, or both. Second, most efforts to pharmacologically activate Type I IFN production result in both tumor-intrinsic and -extrinsic IFN production when applied systemically, resulting in dose-limiting toxicities[28]. Thus, there is a significant need to discover methods to activate tumor-intrinsic Type I IFN signaling in a manner that is independent of the cGAS/STING pathway.

In this study, we trace the fate of mtRNA during apoptosis. We observe that MOMP results in the cytoplasmic release of mtRNA and that executioner caspases-3/7 – also activated by MOMP – prevent mtRNA-mediated activation of a potent antiviral MDA5/MAVS/IRF3 Type I IFN signaling pathway. Thus, a major function of casp3/7 is to prevent inflammation driven by the cytoplasmic release of mtRNA, and by combining apoptosis-inducing cytotoxic or targeted chemotherapies with caspase-3/7 inhibition, it is possible to achieve the long sought-after goal of pharmacologically-induced, tumor-intrinsic, and cGAS/STING-independent Type I IFN production to potentiate anti-tumor immunity.

## Results

### MOMP releases mtdsRNA into the cytosol in a BAX- and BAK-dependent manner

We first generated BAX and BAK (BAX[-/-]BAK1[-/-]) knockout and wild-type controls in BRAF(V600E) mutant A375 melanoma cells with CRISPR/Cas9 technology. To engage apoptosis, control and BAX[-/-]BAK1[-/-] A375 cells were treated with the BRAF inhibitor PLX-4720 and the MCL-1 inhibitor S63845. As previously reported[29], dual BRAF and MCL-1 inhibition promoted MOMP, as evidenced by the cleavage of casp3/7 and their substrate, PARP, an effect that was entirely abrogated in cells lacking BAX and BAK. (Fig. 1a, b and Supplementary Fig. 1a). To determine whether mtRNA is released into the cytoplasm following MOMP, A375 cells were subjected to combination PLX-4720 and S63845 treatment followed by cellular fractionation (Supplementary Fig. 1b). Apoptosis stimulation resulted in a significant accumulation of cytosolic mitochondrial gene mRNA transcripts that were absent in BAX[-/-]BAK1[-/-] cells (Fig. 1c and Supplementary Fig. 1c). Furthermore, staining A375 wild-type and BAX[-/-]BAK1[-/-] cells with the anti-dsRNA (J2) antibody following apoptosis stimulation revealed cytosolic mitochondrial dsRNA species in wild-type cells, but not BAX[-/-]BAK1[-/-] cells, upon PLX-4720 and S63845 treatment (Fig. 1d and Supplementary Fig. 1d). Together, these data extend prior evidence of BAX/BAK-dependent nucleic acid release, and in particular, mtdsRNA release in the setting of PNPase deficiency, by demonstrating that chemotherapy-induced MOMP leads to mtdsRNA release even in the setting of an intact RNA degradosome[4,5,8].

### Caspases-3 and -7 suppress mtRNA-dependent Type I IFN production following MOMP

Given that apoptosis is canonically associated with a lack of inflammatory signaling, we reasoned that a downstream mechanism might prevent mtRNA from activating endogenous antiviral defense pathways during MOMP. Caspases, particularly casp3/7, were originally thought to be vital for cell death following MOMP. However, consistent with the findings of others[30,31], we observed that pan-caspase inhibition does not influence the extent of cell death following MOMP (Supplementary Fig. 1e). Interestingly, executioner caspases have recently been described as essential for the suppression of innate immune signaling during apoptosis, by preventing the recognition of mtDNA and other immunogenic ligands[32]. When MOMP is activated in caspase-deficient settings, a process known as caspase-independent cell death (CICD), antiviral innate immune signaling pathways become activated and secrete Type I IFNs and tumor necrosis factor alpha into the extracellular environment[33–35]. Given their role in preventing mtDNA-dependent inflammation during cell death, we hypothesized that casp3/7 also inhibit mtRNA from activating Type I IFN signaling during apoptosis.

A375 cells stably lacking caspase-3 (CASP3[-/-]), caspase-7 (CASP7[-/-]), or both (CASP3[-/-]7[-/-]) were generated through CRISPR/Cas9 gene deletion and subsequently treated with PLX-4720 and S63845 individually or in combination (Fig. 2a). Induction of MOMP was observed across all the derivative cell lines treated with the drug combination, as demonstrated by the cleavage of caspase-9, while the lack of cleaved PARP in CASP3[-/-]7[-/-] cells undergoing MOMP functionally validated casp3/7 knockout in these cells (Fig. 2a). Interestingly, MOMP led to the phosphorylation of signal transducer and activator of transcription 1 (STAT1), a marker of IFNAR activation[12], in CASP3[-/-]7[-/-] A375 cells (Fig. 2a). In concordance with IFNAR activation, only MOMP-induced CASP3[-/-]7[-/-] A375 cells showed significant transcriptional induction of the ISGs ISG15 and CXCL10 when compared across all conditions (Fig. 2b and Supplementary Fig. 1f)[36]. To identify the cytokine that is activating IFNAR, MOMP was induced in caspase knockout cell lines, and extracellular levels of the Type I IFN family members IFN-α and IFN-β were measured. Whereas CASP3[-/-]7[-/-] A375 cells appreciably secreted IFN-β following MOMP, IFN-α secretion into the extracellular space was not observed (Fig. 2c and Supplementary Fig. 1g, h). Pharmacological induction of CICD was achieved through pairing BRAF and MCL-1 inhibitors with the pan-caspase inhibitor Emricasan in A375 wild-type cells. Treatment with Emricasan appropriately inhibited caspase activity in MOMP-induced A375 cells and led to the activation of Type I IFNs, as demonstrated by loss of cleaved PARP and induction of phosphorylated STAT1, respectively (Fig. 2d). To investigate whether the ligand for Type I IFN signaling originated from the mitochondria, the same three-drug treatment was applied to BAX[-/-]BAK1[-/-] A375 cells. BAX/BAK-null cells entirely prevented the induction of IFIT3 and ISG15 transcripts seen in wild-type cells, indicating the ligand for Type I IFN production is released from the mitochondria in a BAK- and BAX-dependent manner (Fig. 2e). As Type I IFNs are known to be activated following the detection of nucleic acids by innate immune sensors, we postulated that the ligand is a mitochondrial nucleic acid species. Consistent with this hypothesis, CICD induction following selective depletion of both mtDNA and mtRNA using ethidium bromide[37] (Supplementary Fig. 2a–c) blocked the production of ISG transcripts and phosphorylation of STAT1 (Fig. 2f and Supplementary Fig. 2d), suggesting that the relevant ligand was mtDNA, mtRNA, or both. mtDNA primarily activates cGAS/STING in the cytosol or TLR9/MyD88 in endosomes to stimulate Type I IFNs[38]. Interestingly, A375 cells are known to epigenetically silence cGAS through hypermethylation of its promoter, restricting this model's ability to detect cytosolic dsDNA species, including mtDNA (Supplementary Fig. 2e, f)[23]. Additionally, short-hairpin RNA (shRNA)-mediated knockdown of MyD88 yielded no difference in ISG production during CICD in A375

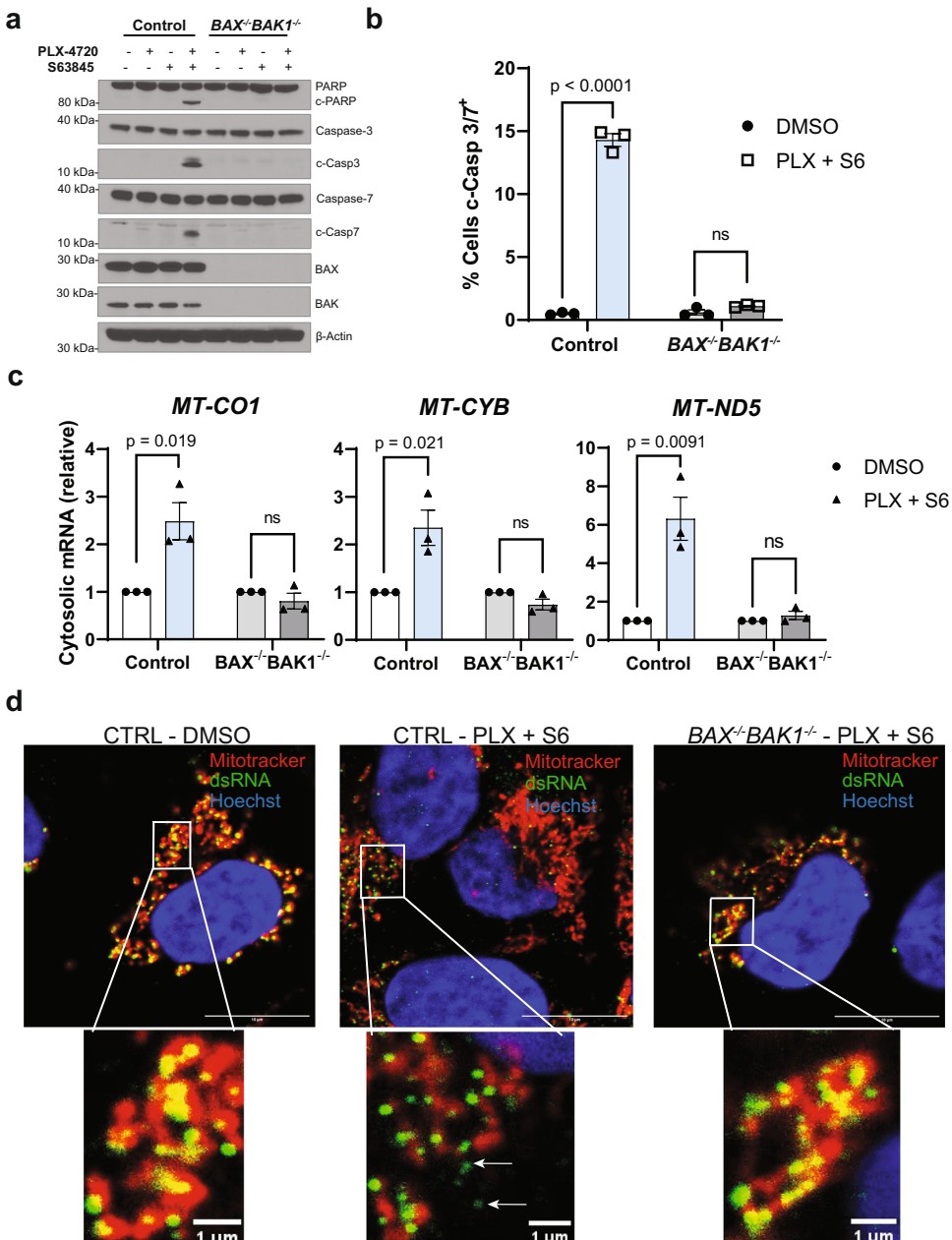

**Fig. 1 | Engagement of mitochondrial outer membrane permeabilization releases mtdsRNA into the cytoplasm in a BAX- and BAK-dependent manner.** A375 *BAX−/−BAK1−/−* or wild-type control cells were treated with 0.5 μM PLX-4720 (PLX), 0.5 μM S63845 (S6), or the combination for 36 h before performing the following experiments. **a** Western blot showing expression of the indicated proteins; representative of three independent experiments (*n* = 3). **b** Flow cytometry measurement of CellEvent™ cleaved casp3/7 FITC positive cells. **c** RT-qPCR of mtRNA from an isolated cytosolic fraction following the specified drug treatment in A375 wild-type or *BAX−/−BAK1* cells. mRNA levels are normalized to each cell line's DMSO treated control. Cytosolic TBP was used as the housekeeping gene. **d** Immunofluorescence images of dsRNA with an anti-dsRNA (J2) antibody were taken. Mitochondria and nuclei are stained with MitoTracker Deep Red and Hoechst, respectively. Scale bars, 10 μm, in original image. Data are representative of two non-overlapping images. Arrows indicate cytosolic dsRNA. **b**, **c** Each dot represents a biological replicate from three independent experiments (*n* = 3). Two-tailed unpaired *t*-test, *p*-values are included in the figure; ns = not significant. Data are presented as mean ± SEM.

cells (Supplementary Fig. 2g). These data suggest mtDNA is likely not contributing to the IFN response in A375 cells during CICD. To uncouple the sensing of mtRNA and mtDNA, A375 cells were treated with a mitochondrial RNA polymerase inhibitor, IMT1B, which specifically depleted levels of mtRNA without affecting levels of mtDNA or the cells' ability to transcribe ISGs upon dsRNA stimulation (Supplementary Fig. 2h, i)[39]. MOMP induction in *CASP3−/−7−/−* A375 cells led to significantly decreased phosphorylation of STAT1 and *IFIT3* and *ISG15* induction in the presence of IMT1B (Fig. 2g, h). Taken together, these data demonstrate that casp3/7 suppress mtRNA-dependent IFN-β production during apoptosis.

## mtRNA activates the cytosolic dsRNA MAVS pathway during CICD

PRRs with the potential to detect dsRNA include the cytosolic RNA helicases melanoma differentiation-associated gene 5 (MDA5), retinoic acid-inducible gene I (RIG-I), and the endosomal toll-like receptor 3 (TLR3)[8,9,40]. MDA5 and RIG-I are cytosolic PRRs that

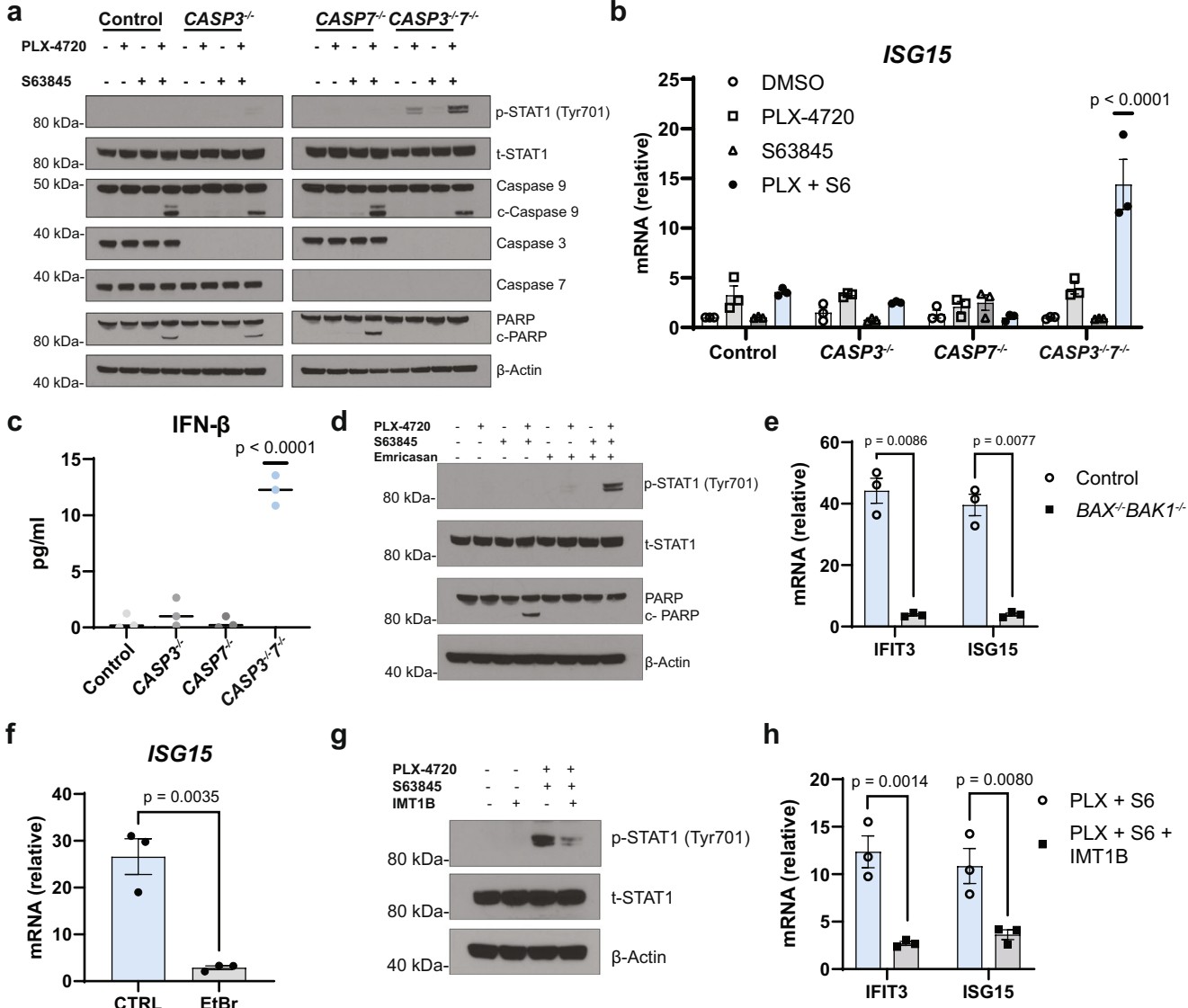

**Fig. 2 | Caspases-3 and -7 inhibit mtRNA-dependent IFN-β production during apoptosis.** A375 wild-type control, *CASP3⁻/⁻*, *CASP7⁻/⁻*, and *CASP3⁻/⁻7⁻/⁻* cells were treated with 0.5 μM PLX-4720 (PLX), 0.5 μM S63845 (S6), or the combination for 24 h before performing **a** western blot for the indicated proteins and **b** RT-qPCR analysis of *ISG15* expression; Two-way ANOVA with Tukey's multiple-comparisons test, *p*-values are included in the figure. Data are presented as mean ± SEM. **c** ELISA analysis for secreted IFN-β collected from conditioned media of A375 wild-type control, *CASP3⁻/⁻*, *CASP7⁻/⁻*, and *CASP3⁻/⁻7⁻/⁻* cells following treatment with 2 μM PLX-4720 and 2 μM S63845 for 24 h; One-way ANOVA with Tukey's multiple-comparison test, *p*-values are included in the figure. Data are presented as mean ± SEM. **d** Western blot for the specified proteins from A375 wild-type lysates treated with the indicated combinations of 0.5 μM PLX-4720, 0.5 μM S63845, and 10 μM Emricasan for 24 h. **e** RT-qPCR analysis for *IFIT3* and *ISG15* expression in wild-type control and *BAX⁻/⁻BAK1⁻/⁻* A375 cells following treatment of 0.5 μM PLX-4720, 0.5 μM S63845, and 10 μM Emricasan for 24 h. **f** A375 wild-type cells were subjected to 5 days of ethidium bromide (EtBr) (100 ng/ml) or control-media pre-treatment. The pre-treated EtBr and control cells were exposed to 0.5 μM PLX-4720, 0.5 μM S63845, and 10 μM Emricasan for 24 h, and RT-qPCR analysis of *ISG15* expression was performed. mRNA levels were normalized to DMSO-treated control cells. **g** A375 *CASP3⁻/⁻7⁻/⁻* cells were treated with the indicated combinations of PLX-4720 (0.5 μM), S63845 (0.5 μM), and IMT1B (2.5 μM) for 24 h and a western blot of the specified proteins was performed. **h** A375 *CASP3⁻/⁻7⁻/⁻* cells were treated with PLX-4720 (0.5 μM) and S63845 (0.5 μM) in the presence or absence of 2.5 μM IMT1B. mRNA levels were normalized to DMSO-treated A375 *CASP3⁻/⁻7⁻/⁻* cells. **a, d, g** The data are a representation of three independent experiments (*n* = 3). **b, c, e, f, h** Each dot represents a biological replicate, and the data are the results of three independent experiments (*n* = 3). **e, f, h** Two-tailed unpaired *t*-test, *p*-values are included in the figure. Data are presented as mean ± SEM. **b, e** mRNA levels are normalized to DMSO-treated control cells.

activate their adapter protein mitochondrial antiviral-signaling protein (MAVS), following ligand engagement. TIR-domain-containing adapter-inducing interferon-β (TRIF) functions as the adapter protein for the endosomal TLR3. Both MAVS and TRIF adapter proteins recruit and activate the kinases TANK-binding kinase 1 (TBK1) and IKK-epsilon (IKKε). Upon recruitment, TBK1 and IKKε phosphorylate the adapter protein, activating a docking site for interferon regulatory factor 3 (IRF3). Once docked, IRF3 is phosphorylated by TBK1 or IKKε, dissociates from the adapter protein, and homodimerizes before translocating into the nucleus, where it facilitates the transcription of IFN-β[41–43].

We hypothesized that either the cytosolic or endosomal dsRNA PRRs control mtRNA-dependent IFN-β production in CICD. To define the PRR(s) and adapter protein(s) involved, *CASP3⁻/⁻7⁻/⁻* A375 cells null for *STING* (*STING1⁻/⁻*), *MDA5* (*IFIH1⁻/⁻*), *RIG-I* (*DDX58⁻/⁻*), *MAVS* (*MAVS⁻/⁻*), or *TLR3* (*TLR3⁻/⁻*) were generated using CRISPR/Cas9. After validating these knockouts at the levels of protein expression and function (Fig. 3a and Supplementary Fig. 3a–c), each cell line was treated with

PLX-4720 and S63845 to induce MOMP. Phosphorylation of STAT1 and production of *ISG15* and *IFIT3* were unchanged from controls in *STING1*[-/-] cells, validating the concept that cytosolic mtDNA does not contribute to Type I IFN production in this setting (Fig. 3a, b). Type I IFN readouts were also insensitive to *TLR3* knockout, suggesting that mtRNA released during MOMP does not activate the endosomal dsRNA pathway (Fig. 3a, b). By contrast, *MDA5*, *RIG-I*, and *MAVS* knockout substantially reduced phosphorylated STAT1 and *ISG15* and *IFIT3* expression (Fig. 3a, b). Specifically, *MDA5* knockout reduced Type I IFN readouts to a greater extent than *RIG-I* knockout (Fig. 3a, b). Together, these data support MDA5 as the primary pattern recognition receptor and MAVS as the adapter protein for mtRNA detection during CICD, with RIG-I playing a less significant role, consistent with previous reports of cytosolic mtRNA sensing[8]. To assess the activation of downstream members of the MAVS signaling pathway, the TBK1/IKKε dual inhibitor Bay-985 was applied to *CASP3*[-/-]*7*[-/-] A375 cells undergoing MOMP, where it blocked the induction of Type I IFN signaling (Fig. 3c and Supplementary Fig. 3d, e)[44]. Finally, to determine the transcription factor downstream of TBK1 and IKKε responsible for mtRNA-dependent IFN-β transcription during CICD, we examined IRF3, which has been previously described to link upstream cytosolic RNA sensing to Type I IFN production[42]. Consistent with this hypothesis, IRF3 phosphorylation was activated in a TBK1/IKKε-dependent manner in casp*3/7*-null A375 cells undergoing MOMP (Fig. 3c). Stable knockout of *IRF3* in wild-type A375 cells blocked Type I IFN production in the setting of treatment with PLX-4720, S63845, and Emricasan without affecting upstream TBK1 phosphorylation (Fig. 3d and Supplementary Fig. 3f). Finally, IMT1B significantly decreased both the phosphorylation of IRF3 and STAT1 in CICD-induced wild-type A375 cells, further implicating mtRNA as the ligand for MAVS pathway activation (Fig. 3e). These data confirm the canonical MAVS pathway controls Type I IFN production following the cytosolic recognition of mtRNA recognition during CICD (Fig. 3f).

### mtRNA is a potent endogenous ligand for Type I IFN production in cGAS/STING-deficient cancer

To investigate whether mtRNA-driven Type I IFN signaling is active in the poorly immunogenic subset of tumors characterized by cGAS/STING-deficiency[25], we selected a panel of *BRAF(V600E)* mutant tumor cell lines that exhibit cGAS-STING signaling defects: cGAS-silenced A375 melanoma cells, STING-silenced SK-MEL-28 melanoma cells, and Colo205 colorectal carcinoma cells that harbor an unknown defect in the signaling axis (Fig. 4a)[23,24]. Each cell line failed to respond to cytosolic dsDNA transfection but transcribed *IFIT3* upon stimulation with a cytosolic dsRNA agonist (Supplementary Fig. 2f and Fig. 4b). Following activation of CICD in SK-MEL-28 and Colo205 cells through combined treatment with PLX-4270, S63845, and Emricasan, we observed a significant increase in *ISG15* and *IFIT3* transcripts that were lost upon ethidium bromide pre-treatment (Fig. 4c). Consistent with the notion that these cells cannot recognize cytosolic DNA, treatment of SK-MEL-28 cells with IMT1B suppressed IFN production following CICD activation, nominating mtRNA as the relevant ligand (Fig. 4d). Similar activation of Type I IFN signaling was observed in STING pathway-deficient cell lines following treatment with the apoptotic agents doxorubicin and paclitaxel in the presence of Emricasan (Fig. 4e, f). As before, this effect was lost in the setting of ethidium bromide pre-treatment (Fig. 4e, f). These chemotherapeutic agents provide an alternative mechanism for the initiation of MOMP and cell death, enforcing the idea that mtRNA-dependent signaling results from the downstream effects of apoptosis rather than the specific mechanisms of a particular drug treatment. These data thus establish CICD as a tool to promote mtRNA-dependent Type I IFN production in the setting of diverse apoptotic agents in STING pathway-deficient tumor cells.

### Tumor-intrinsic mtRNA signaling contributes to CD8[+] T-cell-dependent anti-tumor immunity during CICD

To determine if the impact of mtRNA-driven, tumor cell autonomous Type I IFN production is sufficient to elicit an anti-tumor immune response in vivo, we turned to the classically immunologically cold B16 (*BRAF* wild type) melanoma model[45]. We generated *CASP3*[-/-]*7*[-/-] B16 murine melanoma cells (Supplementary Fig. 4a), then treated these cells (or wild-type controls) with doxorubicin. Gene-Set Enrichment Analysis (GSEA) of RNA-sequencing data from these samples revealed that the most enriched pathway in doxorubicin-treated casp3/7 knockout cells compared to doxorubicin-treated control cells was the Type I IFN response signature (Supplementary Fig. 4b–d). In particular, strong and significant differential expression of the ISGs *ISG15*, *CXCL10*, *IFIT1*, *IFIT3*, and *IFIT3B* was observed (Supplementary Fig. 4b). We validated that *CASP3*[-/-]*7*[-/-] B16 cells undergoing CICD secreted IFN-β and activated the expected Type I IFN signaling markers in secondary assays (Supplementary Fig. 4e–g).

The B16 model maintains intact cGAS/STING and MDA5/MAVS signaling pathways, like other commonly used transplantable syngeneic tumor models. Therefore, we sought to quantify the relative contribution of mtDNA and mtRNA to Type I IFN signaling during CICD in this model. Depletion of all mitochondrial nucleic acids in B16 CASP3[-/-]7[-/-] cells with ethidium bromide entirely suppressed mRNA induction of *CXCL10* (Supplementary Fig. 4h and Fig. 5a). Next, we confirmed the direct contribution of mtRNA to the observed Type I IFN production by applying IMT1B to doxorubicin-treated B16 CASP3[-/-]7[-/-] cells and observing a significant reduction in *CXCL10* and *IFIT3* induction (Supplementary Fig. 4i and Fig. 5b). Finally, we found that knockout of both *STING* and *MAVS* reduced *IFIT3* induction and phosphorylation of TBK1 to similar degrees in doxorubicin-treated B16 cells during CICD (Fig. 5c and Supplementary Fig. 4j). Together, these data suggest that both mtDNA and mtRNA contribute to in vitro Type I IFN production in CICD-induced B16 cells. Accordingly, we find the induction of *IFIT3* mRNA is entirely dependent on IRF3, the transcription factor downstream of both STING and MAVS (Fig. 5d).

To examine the therapeutic impact of mitochondrial nucleic acid sensing during CICD in vivo, we established syngeneic B16 tumors C57BL/6J host mice. While neither doxorubicin nor casp3/7 knockout impaired tumor growth alone, the combination led to a significant impairment in tumor growth (Fig. 5e). This effect was entirely dependent upon tumor cell-intrinsic Type I IFN production, as *IRF3* knockout completely reversed this growth defect (Fig. 5f). Next, we confirmed that loss of *STING* or *MAVS* did not alter tumor growth in B16 *CASP3*[-/-]*7*[-/-] cells (Supplementary Fig. 4k). Interestingly, the growth impairment in B16 cells undergoing CICD was partially dependent on both tumor-intrinsic *STING* and *MAVS* expression (Fig. 5g), suggesting that both tumor-intrinsic mtDNA and mtRNA participate in the Type I IFN-dependent anti-tumor response when the tumor cGAS/STING signaling pathway is intact. Finally, B16 casp3/7-null tumors were treated with saline or doxorubicin in the presence of the following antibodies: control IgG, anti-CD4, anti-CD8α, or anti-PD-1. Anti-tumor responses were lost following CD8[+], but not CD4[+], T-cell depletion and enhanced by PD1 blockade, confirming that tumor cell-derived Type I IFNs lead to a tumoricidal adaptive immune response (Fig. 5h, i). Taken together, these data indicate that both mtDNA and mtRNA are immunogenic ligands capable of activating therapeutic anti-tumor immunity during CICD.

## Discussion

In this study, we demonstrate that engagement of MOMP leads to the release of immunogenic mtRNA species into the cytosol and that the executioner caspases-3 and -7 suppress its cellular detection. Treatment of casp3/7-inhibited tumor cells with various apoptotic agents elicits potent production of IFN-β through the MDA5/MAVS/IRF3 cytosolic dsRNA antiviral signaling axis. Exploiting this signaling axis,

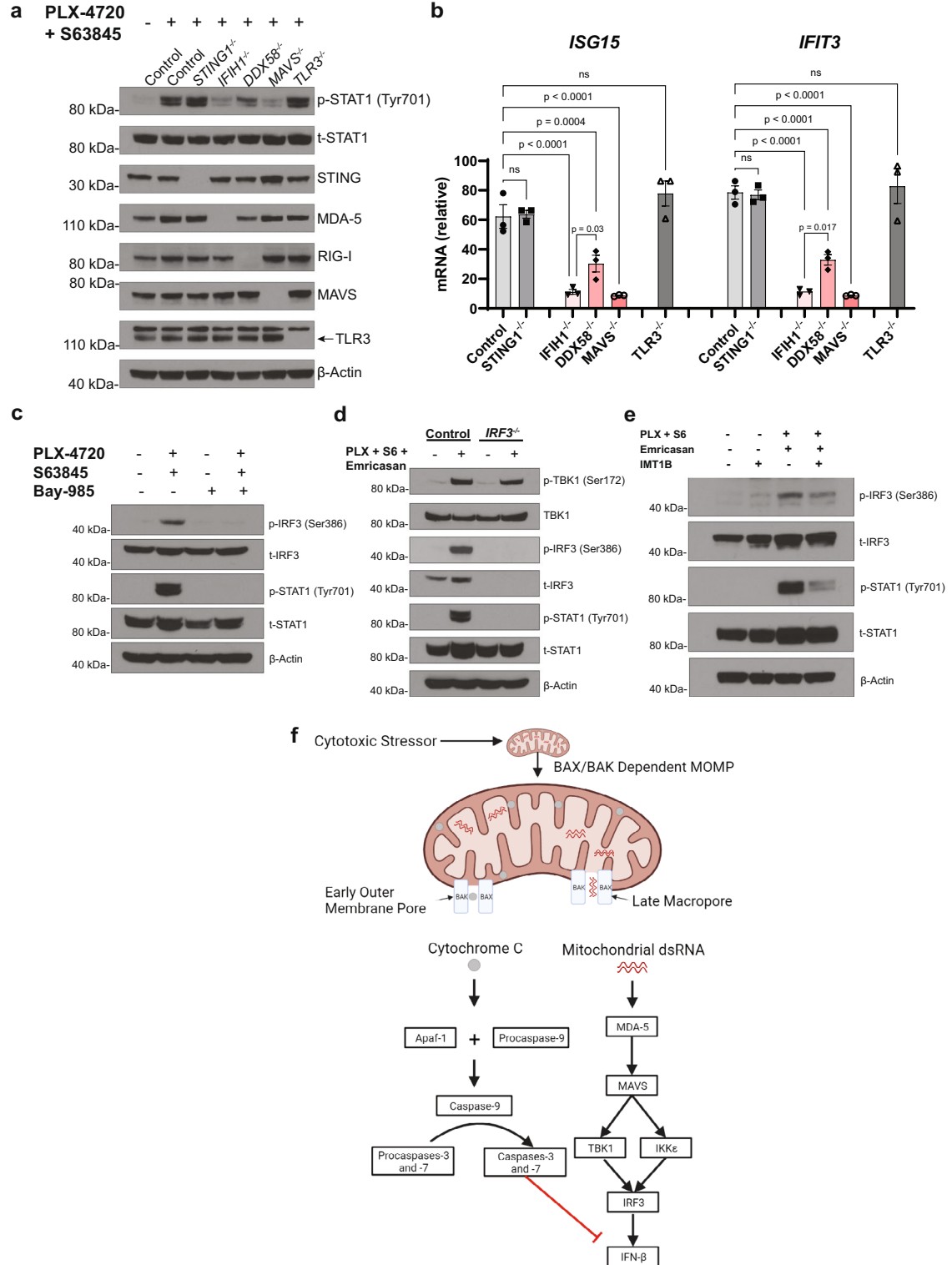

**Fig. 3 | mtRNA activates the MAVS signaling pathway during CICD. a** Western blot taken from A375 *CASP3⁻/⁻7⁻/⁻* cell lysates with an additional CRISPR/Cas9 knockout against the indicated genes. The isogenic cell lines were treated with either DMSO or 0.5 μM PLX-4720 and 0.5 μM S63845 for 24 h. TLR3 protein is shown by the identified band on the blot. The data are representative of two independent experiments (*n* = 2). **b** A375 *CASP3⁻/⁻7⁻/⁻* cells with an additional knockout against the designated gene were treated under identical conditions as in (**a**), and RT-qPCR analysis was performed to measure *ISG15* and *IFIT3* transcript levels. Each dot represents a biological replicate, and the data are the results of three independent experiments (*n* = 3); one-way ANOVA with Tukey's multiple-comparisons test, *p*-values are included in the figure; ns = not significant. Data are presented as mean ± SEM. mRNA levels were normalized to DMSO-treated control cells. **c** A375 *CASP3⁻/⁻7⁻/⁻* cells were treated with PLX-4720 (2 μM) and S63845 (2 μM) in the presence or absence of the TBK1/IKKε inhibitor Bay-985 (0.2 μM) for 24 h and western blots for the indicated proteins were performed on the lysates. **d** Western blot displaying specified proteins from A375 wild-type control or *IRF3⁻/⁻* cells following treatment with 2 μM PLX-4720 (PLX), 2 μM S63845 (S6), and 10 μM Emricasan for 24 h. **e** Western blot of A375 wild-type cells following 24 h treatment with the indicated combination of PLX-4720, S63845, Emricasan, and IMT1B. **f** Proposed mechanism for mtRNA signaling in apoptosis created with BioRender.com. **c**–**e** The data are representative of three independent experiments (*n* = 3).

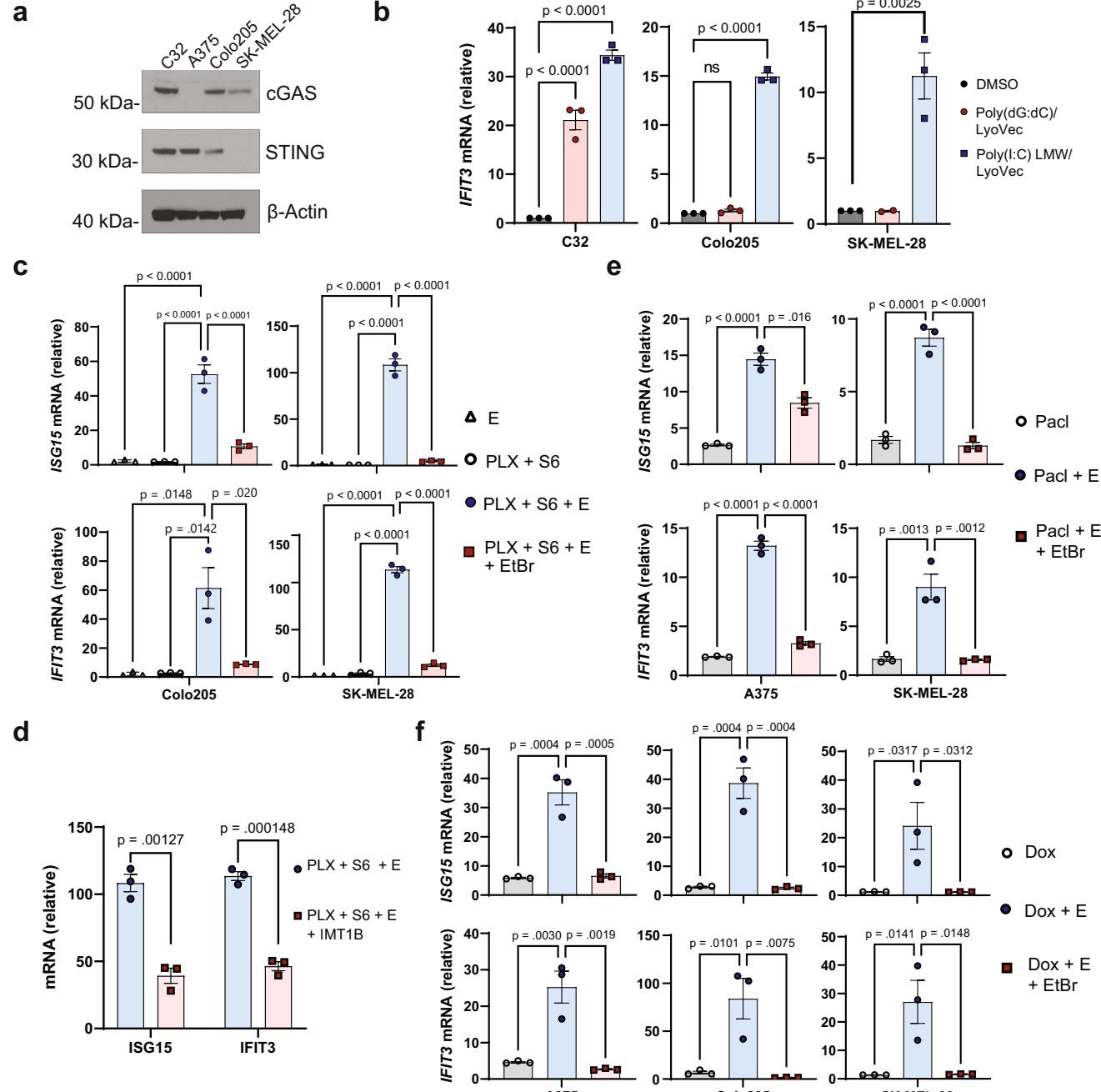

**Fig. 4 | CICD triggers mtRNA-dependent Type I IFN production in cGAS/STING pathway deficient tumor cells. a** Western blot showing C32, A375, Colo205, and SK-MEL-28 baseline expression of indicated proteins. Data are representative of three independent experiments ($n = 3$). **b** Indicated cell lines were treated with 0.5 μg/ml Poly(dG:dC)/LyoVec (cytosolic dsDNA agonist) or Poly(I:C) (LMW)/LyoVec (cytosolic dsRNA agonist) for 16 h. RT-qPCR analysis was performed to measure *IFIT3* expression. Each dot represents a biological replicate, and A375 and Colo205 experiments were performed in biological triplicate ($n = 3$). SKMEL28 dsDNA treatment was performed in duplicate ($n = 2$), and dsRNA treatment was performed in triplicate ($n = 3$). **c** Colo205 and SK-MEL-28 cells were subjected to 5 days of EtBr (100 ng/ml) or control-media pre-treatment. The pre-treated EtBr and control cells were exposed to the designated combinations of 2 μM PLX-4720 and 2 μM S63845 (PLX + S6) and 10 μM Emricasan (E) for 48 h. *ISG15* and *IFIT3* transcripts were measured with RT-qPCR. **d** SK-MEL-28 cells were treated with 2 μM PLX-4720, 2 μM

S63845, and 10 μM Emricasan in the presence or absence of 2.5 μM IMT1B for 36 h. RT-qPCR was performed to measure expression levels of *ISG15* and *IFIT3*. **e, f** A375, Colo205, and SK-MEL-28 cells were conditioned with EtBr or control media as described above. **e** EtBr and control media-treated A375 and SK-MEL-28 cells were subjected to 30 nM and 10 nM paclitaxel (PacI), respectively, with or without Emricasan (E) for 48 h. RT-qPCR was performed to measure expression levels of *ISG15* and *IFIT3*. **f** EtBr and control-treated A375, Colo205, and SK-MEL-28 were treated with 300 nM, 200 nM, and 200 nM doxorubicin (Dox), respectively, with or without 10 μM Emricasan (E) for 48 h. RT-qPCR was performed to measure expression levels of *ISG15* and *IFIT3*. **c–f**, Each dot represents a biological replicate, and the data are the results of three independent experiments ($n = 3$). **b–f** mRNA levels are normalized to DMSO-treated control cells. One-way ANOVA with Tukey's multiple-comparison test, *p*-values are included in the figure; ns = not significant. Data are presented as mean ± SEM.

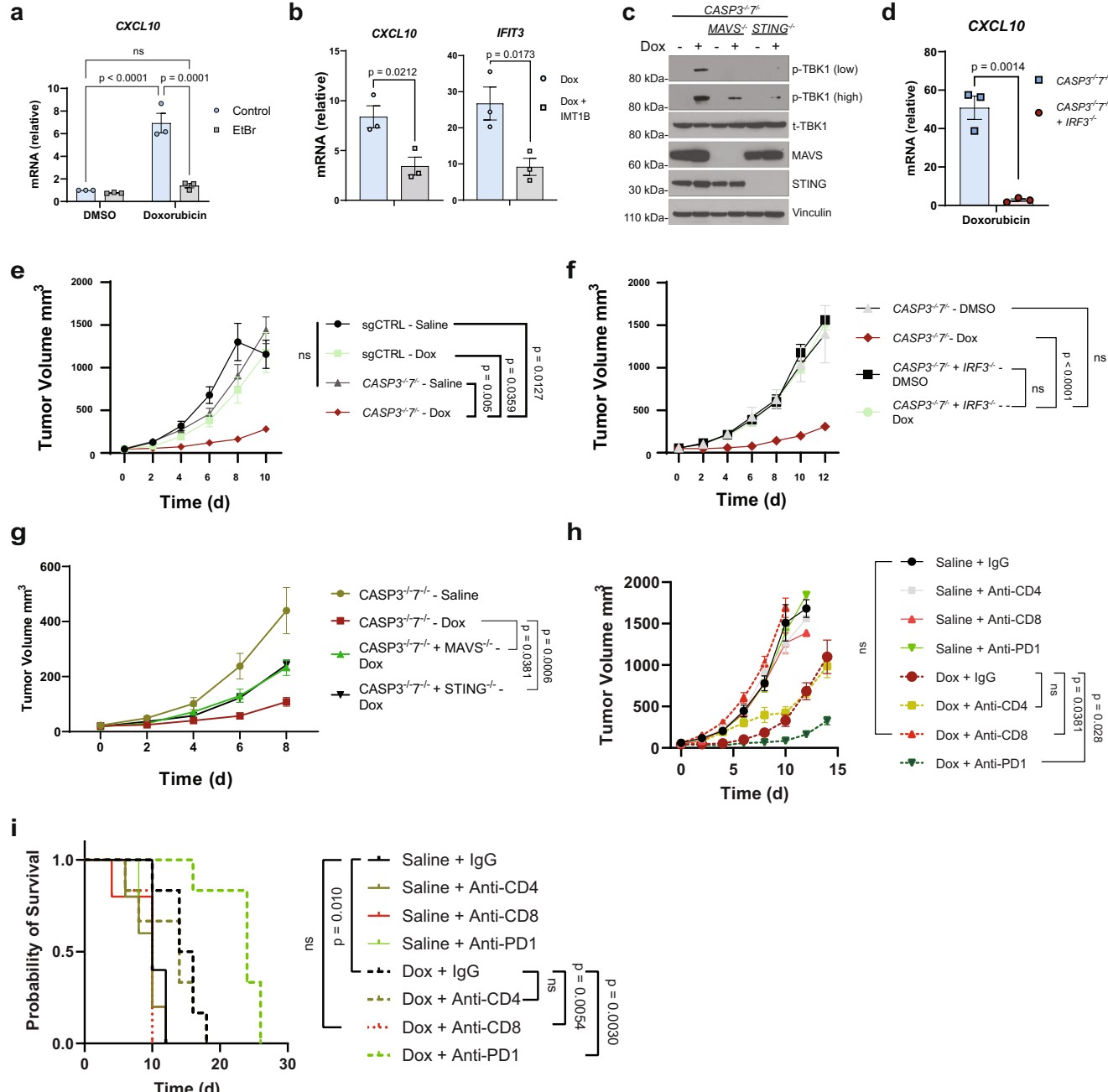

**Fig. 5 | Tumor-intrinsic mtRNA signaling contributes to CD8+-dependent anti-tumor immunity during CICD. a** B16 CASP3⁻/⁻7⁻/⁻ cells were subjected to 5 days of ethidium bromide (EtBr) (100 ng/ml) or control-media pre-treatment before treatment with 1.5 μM doxorubicin for 24 h. RT-qPCR analysis of *CXCL10* expression was performed. **b** B16 *CASP3⁻/⁻7⁻/⁻* cells were treated with 1.5 μM doxorubicin in the presence or absence of 5 μM IMT1B for 24 h before RT-qPCR analysis measuring *CXCL10* and *IFIT3* expression. Two-tailed unpaired *t*-test, *p*-values are included in the figure. Data are presented as mean ± SEM. **c** Western blot performed on B16 CASP3⁻/⁻7⁻/⁻ cells with an additional CRISPR/Cas9 knockout against the specified protein after 24 h treatment with 1.5 μM of doxorubicin. Data are representative of three independent experiments (*n* = 3*)*. **d** RT-qPCR analysis of *CXCL10 expression* in B16 CASP3⁻/⁻7⁻/⁻ cells with or without IRF3 knockout following treatment of 1.5 μM doxorubicin for 24 h. **e** Tumor volume growth curves after subcutaneous injection of B16 wild-type control or *CASP3⁻/⁻7⁻/⁻* cells in the flanks of C57BL/6 mice treated with either doxorubicin (Dox) or saline. All treatment groups had a sample size of seven mice (*n* = 7) except Dox-treated CASP3⁻/⁻7⁻/⁻ tumors, where six mice were used (*n* = 6). **f** Tumor volume growth curves after subcutaneous injection of B16 *CASP3⁻/⁻7⁻/⁻* cells with or without *IRF3* knockout in the flanks of C57BL/6 mice treated with

either doxorubicin (Dox) or saline. *CASP3⁻/⁻7⁻/⁻* tumors had a sample size of five mice (*n* = 5). *CASP3⁻/⁻7⁻/⁻ + IRF3⁻/⁻* tumors had a sample size of six mice (*n* = 6). **g** Tumor volume growth curves after subcutaneous injection of B16 *CASP3⁻/⁻7⁻/⁻* cells with an additional knockout against STING or MAVS in the flanks of C57BL/6 mice treated with either doxorubicin (Dox) or saline. *CASP3⁻/⁻7⁻/⁻* tumors had a sample size of seven mice (*n* = 7). *CASP3⁻/⁻7⁻/⁻ + MAVS⁻/⁻* and *CASP3⁻/⁻7⁻/⁻ + STING⁻/⁻* tumors had a sample size of five mice (*n* = 5). **h** Tumor volume growth curves from mice after subcutaneous injection of B16 *CASP3⁻/⁻7⁻/⁻* cells in the flanks of C57BL/6 mice were treated with either doxorubicin (Dox) or saline in the presence of the indicated antibodies. These results are combined from two independent experiments. All saline treatment groups had a sample size of five mice (*n* = 5). All doxorubicin treatment groups had a sample size of six mice (*n* = 6). (**i**) Kaplan–Meier survival of mice in experiment (**h**). Statistical analysis by two-way ANOVA with Tukey's multiple-comparison test in **e–h** and log-rank Mantel-Cox test in (**i**), *p*-values included in the figure. Data are presented as mean ± SEM; ns = not significant. **a**, **d** One-way ANOVA with Tukey's multiple-comparison test was performed; *p*-values are included in the figure; ns = not significant. Data are presented as mean ± SEM. mRNA levels were normalized to DMSO-treated control B16 cells.

we demonstrated the re-activation of tumor-intrinsic Type I IFN signaling across STING-deficient tumor cell models in vitro. Finally, we established that mtRNA-dependent Type I IFN production provokes CD8[+] T-cell anti-tumor immunity in immune refractory tumors in vivo.

Casp3/7 have previously been identified as playing a key role in preventing STING-dependent IFN signaling during CICD[4,30,33,34]. The data presented in this study broaden these findings, as we show IFN signaling in STING-deficient cancers is also possible through mtRNA's activation of MAVS during CICD. Of note, we confirmed that the models used in this study do not harbor loss-of-function mutations in the RNA degradosome proteins PNPase or SUV3, which would lead to artificially high mtdsRNA species[8]. A375 and SK-MEL-28 cells harbor wild-type SUV3 and PNPase. Colo205 cells contain a T130A mutation in *PNPT1*, but we found no evidence of this variant altering PNPase activity. That said, evidence in other model systems suggesting that STING is the primary driver of IFN signaling during CICD[16,33,34,46] may be driven by the downregulation of the MAVS pathway in those models (a phenomenon thought to be rare in human tumors). Further, the amount of mitochondrial RNA varies from 5 to 30% of total RNA depending on the cell's metabolic requirements[47]. Certain cell types with higher mtRNA synthesis rates or harboring alterations in other as-yet-undefined determinants of mtRNA-MAVS signaling may be differentially primed for MAVS-dependent signaling if CICD is activated.

Our results identify casp3/7 as necessary to restrict mtRNA inflammatory signaling in apoptosis, but the precise molecular mechanism by which casp3/7 inhibit this pathway remains to be elucidated. The Degrabase is a publicly available proteomics database that describes targets of caspase-mediated degradation in various apoptotic contexts[48]. Cross-referencing the Degrabase dataset against the mtRNA signaling pathway identified in this study reveals MAVS and IRF3 as potentially relevant targets for apoptotic caspase cleavage in Jurkat T-cell lymphoma cells. Furthermore, casp3/7 were identified to proteolytically target cGAS, MAVS, and IRF3 during viral infection and apoptosis in THP-1 acute monocytic leukemia cells[49]. Accordingly, we found that both IRF3 and MAVS were cleaved by caspases in apoptotic A375 cells (Supplementary Fig. 5a, b). Interestingly, the expression of caspase-resistant IRF3[D121/I125A] and MAVS[D429/490A] mutants in A375 cells did not activate IFN signaling upon apoptotic stimulation (Supplementary Fig. 5c). These data indicate that although MAVS and IRF3 are cleaved by caspases, they are unlikely to be the only proteins involved in the suppression of mtRNA signaling in apoptosis. One explanation may be that multiple members of the signaling pathway beyond MAVS and IRF3 are being degraded. Alternatively, others have noted that inhibition of casp3/7 may also contribute to IFN production by slowing the kinetics of cellular elimination following MOMP, thus providing sufficient time for mtRNA to be recognized and IFN-β transcription to occur[4,5]. Regardless of the precise mechanism at play, our results establish that genetic or pharmacologic inhibition of caspases-3 and -7 unlocks the inflammatory potential of mtRNA signaling during apoptosis in cancer.

This study reveals an important and foundational role for executioner caspases in maintaining apoptosis as an immunologically silent programmed cell death pathway by suppressing mtRNA inflammatory signaling. Chemotherapy and targeted therapy agents are known to specifically engage apoptosis as a primary mode of tumor cell death[50]. Inhibition of the immunosuppressive functions of casp3/7 enables the activation of mtRNA-dependent viral mimicry in cancer cells undergoing treatment with various cytotoxic chemotherapy agents. We evaluated the therapeutic relevance of mtRNA-dependent Type I IFN signaling in cancer by establishing the signaling axis in two different types of immunologically cold tumor models. First, we established that in human cancer cell lines defective in their ability to signal through the cGAS/STING pathway, mtRNA is a potent agonist capable of reactivating Type I IFN production and consequent immunosurveillance. Second, we established that in vivo tumor models refractory to single agent immune checkpoint inhibition, mtRNA-dependent Type I IFN production was sufficient to activate anti-tumor immunity and restore responsiveness to anti-PD1 therapy. From a translational perspective, these findings provide what is, to our knowledge, the first mechanism-based, systemic pharmacological strategy to induce tumor-specific, cGAS/STING-independent Type I IFN production. Casp3/7 inhibition, when combined with cytotoxic or targeted chemotherapies, may therefore serve as a broadly useful strategy for activating therapeutic anti-tumor immunity or potentiating the activity of complementary immunotherapeutic strategies like checkpoint inhibition.

## Methods

### Study approval
All animal procedures and studies were approved by the Institutional Animal Care and Use Committee at Duke University (IACUC). The IACUC protocol number for the study is A264-19-12.

### Cell lines and reagents
All cell lines were maintained in a humidified incubator at 37 °C with 5% $CO_2$. A375 and Colo205 cells were cultured in RPMI 1640 medium with 10% fetal bovine serum (FBS) and 1% penicillin-streptomycin (P:S). C32 cells were cultured in EMEM with 10% FBS and 1% P:S. A431 and SK-MEL-28 cells were cultured in DMEM with 10% FBS and 1% P:S. B16F10 cells were cultured in DMEM high glucose (4.5 g/l) with 10% FBS and 1% P:S. All cell lines were purchased from the American Type Culture Collection or Duke University Cell Culture Facility. The cells routinely tested negative for mycoplasma contamination and were confirmed to be authentic with STR analysis. PLX-4720 (S1152), S63845 (S8383), Bay-985 (S8935), Emricasan (S7775), and diABZI STING agonist (S8796) were purchased from Selleckchem. IMT1B (HY-137067) was purchased from Medchem. Digitonin (D141) and ethidium bromide (E7637) were purchased from Sigma Aldrich. Poly(I:C) HMW (tlrl-pic), Poly(I:C) (LMW) / LyoVec™ (tlrl-picwlv), and Poly(dG:dC)/LyoVec (tlrl-pgcc) were purchased from Invivogen. For in vitro experimentation, doxorubicin (S1208) was purchased from Selleckchem; for in vivo experiments, DOXOrubicin HCl Injection, USP (Fresenius Kabi) was purchased through the Duke Pharmacy. Anti-mouse CD4 (BE0003-1), anti-mouse CD8α (BE0061), anti-mouse PD-1 (BE0146), rat IgG2b isotype control (BP0090), and rat IgG2a isotype control (BE0089) were purchased through BioXCell. MitoTracker® Deep Red FM (8778) was purchased from Cell Signaling Technology (CST). Anti-dsRNA monoclonal antibody J2 (10010200) was purchased from Exalpha Biologicals. Hoechst 33342 (H3570) and Lipofectamine 3000 (L3000015) were purchased from ThermoFisher (Thermo). MAVS and IRF3 pSIN-3xFlag-IRES-puro plasmids were provided by the Jiang Lab[49].

### Generation of CRISPR/Cas9 knockouts
pSpCas9(BB)-2A-GFP (PX458) was a gift from Feng Zhang (Addgene plasmid # 48138; http://n2t.net/addgene:48138; RRID:Addgene_48138)[51]. sgRNAs were cloned into the PX458 backbone and validated with Sanger sequencing. Plasmids were transfected into cells with lipofectamine 3000 per the manufacturer's guidelines. Seventy-two hours following transfection, the top 10% of GFP+ live cells were sorted into a new population and seeded into a 12-well plate. One week later, the cells were serially diluted and seeded at a concentration of one cell per well in 96-well dishes. Western blots were performed on the clones to confirm knockout. Then, >3 confirmed knockout clones were pooled together in equal amounts to create a knockout population. Wild-type control cells were generated through transfection of an empty px458 plasmid, followed by sorting the top 10% of GFP+ cells and selecting >3 wild-type clones into a new population. All human sgRNA sequences were chosen from the Toronto KnockOut CRISPR Library-Version 3 and are as follows[52]: *BAX*(GATCGAGCAGGGCGAATGGG), *BAK1*(TAAGGTGACCATC TCTGGGT), *CASP3*(AGTTTCTGAATGTTTCCCTG), *CASP7*(GAAGAGGGA CGGTACAAACG), *STING1*(GCAGGCACTCAGCAGAACCA), *IFIH1*(AACTG

CCTGCATGTTCCCGG), *DDX58*(ACTCACCCTCCCTAAACCAG), *MAVS*(TC
TTCAATACCCTTCAGCGG), *TLR3*(GATGCACACAGCATCCCAAA), and
*IRF3*(TTGGAAGCACGGCCTACGGC).

All mouse sgRNA sequences were chosen from the Mouse Toronto KnockOut (mTKO) CRISPR Library and are as follows[53]: *CASP3*
(CATGCAGAAAGACCATACAT), *CASP7*(CGATGATCAGGACTGTGCTG),
*MAVS*(AGGGCAGGATGCTCACCCGG), and *IRF3*(GAACGAGGTTCAGGA
TCCCG).

## Western blot

Cell pellets were resuspended in 1X Cell Lysis Buffer (CST #9803)
supplemented with Peirce Protease and Phosphatase Inhibitor
Mini Tablets (Thermo, #A32959). The solution was agitated by
inversion at 4 °C for 15 min before 4 °C centrifugation at 13,000×*g*
for 10 min. Protein from the resulting supernatant was quantified
by the Bradford assay. The lysates were mixed with NuPage
Sample Buffer (4X) and boiled at 70 °C for 10 min. Samples were
run on NuPage 4–12% Bis-Tris Gels and transferred to PVDF
membranes using the Trans-Blot Turbo system (Bio-Rad). Membranes were blocked for 1 h with 5% milk/PBS-T (w/v) prior to
incubation with primary antibodies overnight at 4 °C. Primary
antibodies were diluted 1:1000 in 5% bovine serum albumin
(BSA)/PBS-T (w/v) and were purchased from Cell Signaling as
follows: PARP (#9532), Cleaved Caspase-3 (Asp175) (#9661),
Cleaved Caspase-7 (Asp198), BAX (#2772), BAK (#12105), β-Actin
(#4970), Calreticulin (#12238), COX IV (#4844), Vinculin
(#13901S), phospho-Stat1 (Tyr701) (#9167), Stat1 (#9172),
Caspase-9 (#9502), Caspase-3 (#9662), Caspase-7 (#12827), STING
(#13647), MDA-5 (#5321), RIG-I (#3743), MAVS (#3993), MAVS
(#83000), Toll-like Receptor 3 (#6961), Phospho-IRF-3 (Ser386)
(#37829), IRF-3 (#4302), Phospho-TBK1/NAK (Ser172) (#5483),
TBK1/NAK (#3504), cGAS (#79978), and Cleaved PARP (#9548).
Monoclonal ANTI-FLAG® M2 antibody (F3165) was purchased
from Sigma. Following primary antibody incubation, membranes
were washed with PBS-T prior to incubation at room temperature
with a species-specific HRP conjugated secondary antibody (CST,
#7076s or #7074s) at a dilution of 1:5000 in 5% milk/PBS-t (w/v)
for 1 h. Membranes were again washed with PBS-T before developing with Pierce ECL (Thermo, #32106) or SuperSignal West Pico
chemiluminescent substrate (Thermo, #34578) and imaging.

## Caspase-3/7 activation assay

Cells were stained with CellEvent™ caspase-3/7 green flow cytometry
assay kit (Thermo, #C10427) according to the manufacturer's protocol. Data were acquired with a FACSCanto II (BD Biosciences) flow
cytometer, and the results were analyzed in Flowjo (TreeStar).

## Cytosolic RNA extraction

Cytosolic extracts were isolated largely as previously described[54].
Briefly, cells were resuspended in 500 µl of a buffer containing 150 mM
NaCl, 50 mM HEPES, and 40 µg/ml of Digitonin before being agitated at
4 °C for 15 min. The homogenates were spun down at 2000×g for 5 min,
and the supernatant containing the cytosolic fraction was transferred to
a fresh tube. The cellular pellet from this spin was saved for western blot
analysis. The cytosolic fraction was purified through three more 2000×*g*
centrifugations to ensure no contamination of nuclear, membrane, or
mitochondrial proteins. The purity of the cytosolic and pellet fractions
was confirmed through western blotting. Total RNA was extracted from
our cytosolic fraction using an RNeasy Plus RNA extraction kit (Qiagen).
A DNase I digest was performed on our cytosolic RNA extract to remove
any remaining mtDNA before further processing.

## Quantitative PCR

For RT-qPCR, total cellular RNA was extracted from conditioned cell
pellets using an RNeasy Plus RNA extraction kit (Qiagen). One

microgram of purified RNA was normalized across all conditions,
and cDNA was generated using iScript cDNA Synthesis kit (Bio-Rad, #
1708891). The cDNA product was diluted 1:4 and subjected to
qPCR with TaqMan™ Fast Advanced Master Mix, no UNG, the housekeeping VIC-MGB probe *TBP* (Hs00427620_m1 or Mm01277042_m1),
and one of the following FAM-MGB probes: *MT-CO1* (Hs02596864_g1),
*MT-CO2* (Hs02596865_g1), *MT-ND5* (Hs02596878_g1), *MT-ND6*
(Hs02596879_g1), *MT-CYB* (Hs02596867_s1), *MT-ATP8*
(Hs02596863_g1), *ISG15* (Hs01921425_s1), *IFIT3* (Hs01922752_s1),
*CXCL10* (Hs00171042_m1), *IFIT3* (Mm01704846_s1), *CXCL10*
(Mm00445235_m1), *CYTB* (Mm04225271_g1), *COX1* (Mm04225243_g1),
and *ND5* (Mm04225315_s1). The reaction mixtures were conducted
according to the manufacturer's instructions on a CFX384 Touch RealTime PCR Detection System. Fold expression was calculated by normalizing cycle threshold (Cq) values to the TBP reference gene and
normalizing samples to the control sample in accordance with the $^{\Delta\Delta}$Cq
method. For mtDNA measurements, total DNA was extracted from
conditioned cell pellets with the DNeasy Blood & Tissue Kit (Qiagen);
100 ng of purified cellular DNA was included in a reaction mixture of
TaqMan™ Fast Advanced Master Mix, the nuclear DNA probe *TBP*, and
one of the mitochondrial DNA probes COX1, ND5, or CYB. The TaqMan
probes, reaction conditions, and analysis were identical to our RT-
qPCR protocol described above.

## ELISA

Cell culture supernatants were collected following drug treatment
and spun down at 1200 RPM for 5 min before being transferred to a
fresh 1.5 ml Eppendorf. The concentration of Human IFN-β in the
supernatant was determined with a Quantikine ELISA Kit (Novusbio,
#DIFNB0). The concentration of mouse IFN-β was determined with a
Quantikine ELISA Kit (R&D systems, #MIFNB0). Human IFN-α
supernatant concentrations were determined with an All-Subtype
ELISA Kit, High Sensitivity (PBL Assay Science, 41135-1). All ELISA
protocols were conducted according to the manufacturer's
instructions.

## Mitochondria DNA depletion

Cells were cultured in media containing 10% FBS, 1% penicillin-streptomycin, 1 mM sodium pyruvate, 4.5 mg/ml of glucose, 50 µg/ml of
uridine, and 100 µg/ml of ethidium bromide for 5 days, as previously
described[37]. Control cells were grown in parallel culture conditions but
were free of ethidium bromide exposure. mtDNA depletion was confirmed through quantitative PCR.

## RNA-seq gene expression analysis

**Experimental conditions.** B16 wild-type control or *CASP3*[-/-]*7*[-/-] cells
were treated for 72 h with DMSO or 0.5 µM doxorubicin in biologically
independent triplicates. RNA was isolated from whole cells with
RNEasy Mini kit (Qiagen) and sent for paired-end non-stranded RNA-
seq by Novogene.

**Data processing.** Raw fastq files were trimmed using TrimGalore followed by quality control analysis with FastQC. Reads were mapped to
the mouse genome (GRCm38) using the HISAT2 alignment tool[55].
Reads were filtered to include only those mapping to a single genomic
locus, and read counts were compiled with featureCounts[56]. Normalization and differential expression analyses were conducted with the
DESeq2 package in the R programming environment[57].

**Gene set enrichment analysis.** Gene list outputs from differential
expression data analysis from RNA-seq experiments were sorted by
statistical rank and imported into GSEA software (v4.1.0, The Broad
Institute)[58]. The pre-ranked gene list was processed with default settings and size filters for analysis within the Hallmark collection of gene
signatures.

### Short-term cell viability assay (GI-50)

A375 cells were seeded at a density of 1000 cells per well in 96-well plates and left to adhere overnight. Then, 24 h following seeding, the drug was added at the specified concentration. Cell titer Glo (Promega) was used to quantify viability 3 days after the addition of the drug. The relative cell viability was determined by normalizing the raw luminescence values for each treatment condition to DMSO-treated wells. For experiments involving two drugs, one drug was kept at a constant concentration across all wells, and a serial dilution of a second drug was added on top of the background drug, with relative cell viability being normalized to the luminescent readout of the background drug only. Dose-response curves were fit using GraphPad/Prism 7/8 software. GI-50 values were interpolated from the resultant graphs as the dose corresponding to 50% cell viability relative to DMSO-treated cells.

### Short-hairpin RNA (shRNA) knockdown

Lentivirus was generated as previously described[59]. A375 cells were stably transduced with lentivirus containing Tet-pLKO-puro (Addgene #21916) plasmids with shRNA sequences created from the TRC shRNA library. The cells were selected for plasmid integration by treatment with puromycin (2 μg/ml) for 48 h. The knockdown efficiency of each shRNA was confirmed by treating the cells with 100 ng/ml of doxycycline for 48 h and immunoblotting for the protein of interest. shRNA sequences are as follows: Scramble (caacaagatgaagagcaccaa), MyD88 (gcagagcaaggaatgtgactt).

### Immunofluorescence

Mouse anti double-stranded RNA J2 (exalpha 10010500) Immunofluorescence images were generated largely as previously described[8] with slight modifications accounting for experimental conditions. A375 cells were seeded on 18-mm coverslips in wells of a 24-well plate at 20,000 cells per well and incubated in a humidified incubator at 37 °C with 5% overnight. The following day, the cells were treated with 0.5 μM PLX-4720 (PLX), 0.5 μM S63845 (S6), or the combination for 36 h. One hour prior to fixation, 200 nM of MitoTracker Deep Red was added to the culture media. Culture media from each well was then aspirated, and cells were washed one time with 1X PBS followed by fixation and permeabilization in 4% paraformaldehyde (v/v), 0.25% Triton X-100 (v/v), and 2 μg/ml Hoechst 33342 stain in PBS for 30 min at room temperature. After dual fixation and permeabilization, cells were washed with 1X PBS three times, followed by incubation in the presence of the anti-J2 primary antibody (2.5 μg/ml) diluted in 3% BSA/PBS (w/v) overnight at 4 °C. Cells were again washed with PBS three times and incubated in secondary goat IgG anti-mouse conjugated with Alexa Fluor 488 (Thermo, #A-11001) at 2 μg/ml for 1 h. Following secondary staining, each well was washed an additional three times with 1X PBS, and coverslips were gently removed from each well and mounted with Dako fluorescence mounting medium (Agilent, S3023) onto Superfrost Plus microscopy slides (VWR, 48311-703) and stored protected from light at 4 °C until imaging. Fixed and stained cells were imaged using a 20X objective on a Leica SP5 inverted confocal microscope. A slide containing unstained cells was used as a negative control, and a minimum of at least two images from non-overlapping fields of view were taken for each condition. All image analysis was performed using FIJI/ImageJ software.

### In vivo studies

All cell lines were confirmed to be mycoplasma negative before injection in vivo. Cells were resuspended in sterile PBS at a concentration of $5 \times 10^6$ cells/ml, and 100 μl of the suspension was injected into the flank of ~6- to 8-week-old female C57BL/6 mice. Tumor size was measured every 48 h with calipers, and tumor volume was calculated by the formula: $V = (L \times W \times W)/2$ ($L$ = longest diameter and $W$ = shortest diameter). When tumor volume reached ~50 mm³, mice were randomized into treatment groups; 100 μl of DOXOrubicin HCl Injection, USP, or sterile saline was injected intraperitoneally into each mouse. Doxorubicin was administered at a dose of 5 mg/kg weekly. Anti-mouse CD4, anti-mouse CD8α, anti-mouse PD-1, and their isotype control antibodies were intraperitoneally injected at 200 μg per mouse. The first antibody dose was administered 1 day prior to starting saline or doxorubicin treatment and continued at a bi-weekly interval for the remaining course of the study. Tumor size was monitored every 48 h until IACUC-approved endpoints, including when tumors reached ~1500 mm³, tumors were ulcerated, or the mice displayed clinical impairments.

### Statistical analysis

All results are shown as means ± SEM unless otherwise shown. $p$-values were determined using unpaired, two-tailed Student's $t$-tests, Mann–Whitney test, or, for grouped analyses, one-way or two-way analysis of variance (ANOVA) with Tukey's post hoc test; $p < 0.05$ was considered statistically significant. Unless otherwise noted, all experiments were performed a minimum of three times, and measurements were taken from individual biological replicate samples.

### Reporting summary

Further information on research design is available in the Nature Portfolio Reporting Summary linked to this article.

## Data availability

All data associated with this study are available in the main text or the supplementary materials. RNA-seq data from B16 sgCTRL and sgCasp3/7 cells treated with doxorubicin is available at Gene Expression Omnibus (GEO) (Accession Number GSE210377, https://www.ncbi.nlm.nih.gov/geo/query/acc.cgi). Source data are provided with this paper.

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

## Acknowledgements

This research was supported by Duke University School of Medicine start-up funds and support from the Duke Cancer Institute (K.C.W.), grant support from an anonymous donor to the Duke Cancer Institute (K.C.W.), a gracious donation from the Nancy Genovese Fund—Fighting Ovarian Cancer Fund (K.C.W.), National Institutes of Health awards R01CA207083 and R01CA263593 (K.C.W.), F30CA206348 (K.H.L.), K00CA245732 (J.P.H.), and F32CA206234 (R.S.S.), the Duke Undergraduate Research Support Office (R.W.), and the Duke Medical Scientist Training Program (T32 GM007171 to S.T.K. and K.H.L.). We would like to thank Dr. Zhengfan Jiang for graciously providing us with caspase-resistant IRF3 and MAVS mutant plasmids. We also would like to thank Dr. Bin Li at the Duke Cancer Institute Flow Cytometry Core for his continual help with cell sorting.

## Author contributions

S.T.K. and K.C.W. conceptualized the project. S.T.K., J.P.H., K.H.L., and K.C.W. were responsible for methodology. Mechanistic and validation studies were performed by S.T.K., R.W., R.S.S., J.L., and J.P.H. Data were curated by S.T.K. and K.C.W. S.T.K. and R.W. were responsible for visualization. The original draft was written by S.T.K., R.W., and K.C.W. All authors reviewed and edited the paper. K.C.W. supervised the project.

## Competing interests

K.C.W. is a founder, consultant, and equity holder at Tavros Therapeutics and Celldom and has performed consulting work for Guidepoint Global, Bantam Pharmaceuticals, and Apple Tree Partners. The remaining authors declare no competing interests.
