## [Peer Review File · Nature Communications]

Reviewers' Comments:

Reviewer #1:

Remarks to the Author:

In this study Killarney and colleagues investigate a role for mtRNA in immunogenic cell death (MOMP induced apoptosis under caspase-inhibited conditions – CICD). They nicely show that mtRNA is released from mitochondria following MOMP, prior to a series of experiments demonstrating that in cGAS-STING deficient cells mtRNA can elicit an interferon response, contributing to anti-tumor immunity in a syngeneic tumour model – as the authors discuss, targeting this pathway may be particularly relevant in STING deficient tumours. The study is timely, well conducted and the results in the main, support the authors' model. I have a few comments that should be addressed, experimentally or textually.

- mt dsRNA is thought to be very unstable, subject to constant degradation in the mitochondria, the authors' model suggests not (given that MDA5 is a sensor of dsRNA). Thus, is the mtRNA being assayed in the relevant cell models ds or ss mtRNA? – one feasible approach to measure this would be to immunostain control/MOMP cells with J2 ab specific to dsRNA (see PMID: 30046113).

- use of the recently described mtRNA synthesis inhibitor IMT1B is a powerful approach to demonstrate a role for mtRNA transcription/mtRNA (independent of mtDNA levels), Fig 2h. An important negative control to include here is to assay whether IMT1B affects inflammation (IFIT3, ISG3 levels), induced by non-mtRNA activators of MDA5/MAVS, and cGAS-STING (e.g. through transfection of dsRNA or DNA respectively). Use of the inhibitor should be extended to figures 3c and d, to determine a role for mtRNA transcription on the activation of TBK1/IRF3/pSTAT signalling seen herein.

- point of discussion, many labs Kile, Flavell, Tait, Fu have shown loss of STING effectively prevents an interferon response post-MOMP, this is in contrast to the findings of the current study (i.e. MDA5/MAVS drives IFN in the absence of STING). While I am not suggesting the authors decipher the reasons behind this, I do think it should be raised in the discussion – with potential speculation as to underlying reasons (possibly variable degradation of mt dsRNA between cell types could be one).

- the authors state line 43, "the mitochondrial genome results in long, double-stranded RNA (dsRNA) species within the mitochondrial matrix^{8,9}, its localization and potential to activate cytosolic nucleic acid sensing pathways following MOMP has never, to our knowledge, been investigated." – others have previously shown that BAX/BAK silencing prevents mitochondrial mt dsRNA release, implicating MOMP in this process (see PMID: 30046113), thus the authors should amend this statement.

Reviewer #2:

Remarks to the Author:

In this manuscript, Killarney et al present a model where the pro-apoptotic executioner caspases inhibit mtRNA mediated activation of MAVS induced IRF3 and IFN β activation. They provide data to suggest this as a potential mechanism to target cancers where the cGAS/STING pathway is impaired by combining conventional chemotherapy with caspase inhibition. Altogether the data is strong and supports the conclusions of the paper. My main concerns are 1. whether the data strongly convinces that it is RNA released from mitochondria that triggers this pathway and 2. The lack of a mechanism of how caspase-3 and -7 inhibit the IRF3 pathway. Specifically:

Major points

1. To demonstrate that the activation of the IFN pathway in the absence of caspases is due to RNA release from the mitochondria, the authors a) deplete DNA with EtBr and b) block mtRNA production with IMT1B, an inhibitor of mitochondrial RNA polymerase. While the genes in the pathway of interest are not expressed under these conditions (Fig 2f-h, Extended data Fig 2 c and f). However, these experiments lack controls for specificity. If DNA is depleted, then it would be expected that all expression would be impaired so measuring transcript levels does not address whether DNA is contributing to this pathway. Similarly, if RNA is depleted. The specificity of IMT1B

for mitochondrial RNA should be demonstrated by including a positive control showing that nuclear transcription of genes not in this pathway is not impacted and stronger evidence is needed to show that mtDNA does not contribute in any way.

2. One of the main conclusions from the paper is that mtRNA and not mtDNA is the trigger for IRF3 signaling in the absence of caspases. However, apart from the experiments described in the above comment, there is not much evidence to support this. Ideally, an invitro experiment where the supernatant collected from isolated mitochondria treated with the drug cocktail (plus or minus IMTB1) is shown to trigger the IRF3 pathway in caspase-3/7 DKO extracts would answer this question but I am not certain if activation of this pathway can be measured in cell free extracts. This or a similar experiment should be considered and, at the very least, discussed.

3. The authors acknowledge that they do not know the targets of caspase-3 and-7 in this pathway but speculate that they could be MAVS and IRF3 based on their inclusion in the Degradase of caspase targets. Given that they have two very appropriate candidates, the authors should include an experiment to rule in or out these proteins as caspase substrates in this pathway.

4. Figure 2c. The amount of IFNbeta released is 15pg/ml. This seems very small. The authors should include a positive control (a known inducer of IFNbeta) for comparison to show that this is a biologically relevant amount.

Minor points

1. Casp3^{-/-} and casp7^{-/-} should be CASP as they are human genes.

2. Western blots (e.g Figure 1A) should show the full length caspases and not just the cleaved fragment.

3. The colors of the lines and symbols in Figure 5 d-g make it difficult to distinguish the different lines (especially in g where there are no symbols). The blue, dark blue and black lines look too similar, as do the orange, dark orange and red.

Reviewer #3:

Remarks to the Author:

A manuscript by Killarney et al. describes a series of experiments that aim to reveal the role of caspase 3/7 in the prevention of mt-dsRNA-driven Type I interferon response. The Authors confirmed and further explored previous observations that:

(i) release of dsRNA from the mitochondrial matrix is BAX dependent (Dhir et al. Nature 2018)

(ii) MDA5 is the primary sensing receptor of the mitochondrial dsRNA (Dhir et al. Nature 2018)

(iii) mitochondrial nucleic acids are released by herniation of the inner mitochondrial membrane (McArthur et al., Science 2018)

(iv) caspases can prevent mitochondria-stimulated Type I interferon (Rongvaux, A. et al. Cell 2014).

Still, the manuscript describes important advances in the field. This comprehensive and well-designed study summarises the cascade of events from mt-dsRNA release to interferon β expression. Importantly, the Authors conclude that caspase 3/7 prevents activation of Type I interferon induction under mt-dsRNA release. Moreover, they show that inhibition of caspases combined with mt-dsRNA release from mitochondria can be used as an anti-cancer strategy. Overall, this is a well-written manuscript that describes technically sound experiments. Conclusions are well supported, and the manuscript is of general interest. I recommend its acceptance for publication in Nature Communications. I have one comment that Authors should modify their text to indicate that some of their observations have already been done by others. Otherwise, a reader of the manuscript can be misled. However, the manuscript still shows a high level of novelty.

Lanes 95-96: BAX-dependent mt-dsRNA release was described by Dhir et al. Nature 2018

Lanes 177-179: MDA5 as the primary pattern recognition receptor of mt-dsRNA was identified by Dhir et al. Nature 2018

Lanes 85-96: original idea of the role of the herniation of the inner mitochondrial membrane into the cytoplasm comes from McArthur et al., Science 2018

Lane 278-280: possible role of caspases in the regulation of mt-dsRNA triggered Type I interferon can be deduced from Rongvaux, A. et al. Cell 2014; this should be indicated in the text

Reviewer #4:

Remarks to the Author:

Killarney et al. reported that the leaked mtRNA during apoptosis activated type I IFN production. The authors found that chemotherapy-induced apoptosis induced IFN in tumors with defective cGAS/STING pathway. Using knockout cells, the authors found that mitochondrial double-stranded RNA activated IFN through MDA5, however, this finding is a recapitulation of previous work published in Nature (doi: 10.1038/s41586-018-0363-0) in 2018. They further found that mtRNA activation might be involved in anti-tumor activity using xenograft mouse models. Although the story integrates MOMP mtRNA leakage with mtRNA sensing and IFN activation, unfortunately, the novelty is moderate at the standing point of the reviewer.

Regarding experiments, there are several questions:

1. It seems that only three mtRNA were examined by RT-qPCR for the mtRNA leakage assays. There are >30 mtRNA genes. What is the rationale to cherry pick these two gene?
2. In fig2b, the ISG15 mRNA level in control cells is very low; however, there is about 60-fold of ISG15 in the control cells in fig 3b. According to the data, wild type cells had a low IFN response due to the caspases. But in reality, most tumors have intact caspases. How do the author interpret it immunotherapy?
3. What is mechanism of how caspase3/7 inhibits IFN? There are several papers that have shown that caspases can inhibits IRF3 signaling. Previous work should be discussed.
4. B16 cells have the functional cGAS/STING pathway. The xenograft experiments cannot exclude the effects of mtDNA, which might play a major role.
5. Many papers about mtRNA are not discussed.

Point-by-point responses to reviewers

We thank the reviewers for their positive appraisals of our work and their detailed and constructive questions. In the text below, we provide point-by-point responses to each question raised (**bold text**), pointing the reviewers to updates we have made to the manuscript in response to their comments. These updates have significantly clarified our manuscript's presentation and strengthened our conclusions.

Reviewer #1 (Remarks to the Author):

In this study Killarney and colleagues investigate a role for mtRNA in immunogenic cell death (MOMP induced apoptosis under caspase-inhibited conditions – CICD). They nicely show that mtRNA is released from mitochondria following MOMP, prior to a series of experiments. demonstrating that in cGAS-STING deficient cells mtRNA can elicit an interferon response, contributing to anti-tumor immunity in a syngeneic tumour model – as the authors discuss, targeting this pathway may be particularly relevant in STING deficient tumours. The study is timely, well conducted and the results in the main, support the authors' model. I have a few comments that should be addressed, experimentally or textually.

*We thank the reviewer for their thoughtful feedback on our work. In particular, we appreciate the insightful comments regarding the discernment of single vs double-stranded mitochondrial RNA (mtRNA) in the activation of the proposed pathway. We also agree with the reviewer's recommendation that we should address previous reports identifying STING as the sole pathway controlling IFN signaling during caspase-independent cell death (CICD). In response to these comments, we have performed several new experiments which serve to address these questions and, in doing so, undoubtedly further our understanding of the biology exposed in this work. We have amended the text to reflect these changes. In each of the below responses, we highlight new figures/figure panels and other content added during revision using **red font**.*

Major Points:

1. mtDNA is thought to be very unstable, subject to constant degradation in the mitochondria, the authors' model suggests not (given that MDA5 is a sensor of dsRNA). Thus, is the mtRNA being assayed in the relevant cell models ds or ss mtRNA? – one feasible approach to measure this would be to immunostain control/MOMP cells with J2 ab specific to dsRNA (see PMID: 30046113).
 - **We thank the reviewer for their keen observation that our data do not distinguish between single- and double-stranded mtRNA in the context of MAVS pathway activation. As the reviewer mentions, the relevant pattern recognition receptor, MDA5, is known to only recognize long double-stranded RNA (dsRNA) species, which implies that mtDNA would be the relevant ligand in this context. Additionally, previous work specifically identified mtDNA as the ligand for MDA5 activation in human cells (Dhir et al., 2020 Nature). To experimentally address this point, we performed the reviewer's recommended immunofluorescence experiment (**Fig. 1d and Extended Fig. 1c**). Importantly, we observed a BAX- and BAK-dependent increase in cytoplasmic mtDNA upon PLX-4720 and S63845 combination therapy.**
2. Use of the recently described mtRNA synthesis inhibitor IMT1B is a powerful approach to demonstrate a role for mtRNA transcription/mtRNA (independent of mtDNA levels), Fig 2h. An important negative control to include here is to assay whether IMT1B affects inflammation (IFIT3, ISG3 levels), induced by non-mtRNA activators of MDA5/MAVS, and cGAS-STING (e.g. through transfection of dsRNA or DNA respectively). Use of the inhibitor should be extended to figures 3c and d, to determine a role for mtRNA transcription on the activation of TBK1/IRF3/pSTAT signalling seen herein.

- We appreciate the reviewer's helpful comments regarding our IMT1B dataset. A negative control condition demonstrating that IMT1B does not influence mtRNA-independent IFN production would be a helpful addition to our study. To this end, we treated A375 cells with a dsRNA agonist, (Poly (I:C)), in the presence or absence of IMT1B and found no difference in *IFIT3* or *ISG15* mRNA induction (Extended Fig. 2i). Additionally, we expanded our IMT1B data to implicate the role of mtRNA transcription in activating our proposed signaling pathway with immunoblots. We found that IMT1B reduced phosphorylation of STAT1 in A375 *CASP3*^{-/-} cells undergoing PLX-4720 and S63845 treatment (Fig. 2g). Furthermore, A375 wild-type cells treated with PLX-4720, S63845, and Emricasan showed substantial reduction of p-STAT1 and p-IRF3 in the presence of IMT1B (Fig. 3e).
3. Point of discussion, many labs Kile, Flavell, Tait, Fu have shown loss of STING effectively prevents an interferon response post-MOMP, this is in contrast to the findings of the current study (i.e. MDA5/MAVS drives IFN in the absence of STING). While I am not suggesting the authors decipher the reasons behind this, I do think it should be raised in the discussion – with potential speculation as to underlying reasons (possibly variable degradation of mt dsRNA between cell types could be one).
 - We thank the reviewer for their insightful comment summarizing prior work surrounding STING-dependent IFN production in CICD. Based on these previous findings, we were surprised to see IFN signaling across STING-deficient cancers, and we felt this finding warranted further investigation. We included a new section in our discussion addressing this topic. (Lines 298-312)
 4. The authors state line 43, “the mitochondrial genome results in long, double-stranded RNA (dsRNA) species within the mitochondrial matrix^{8,9}, its localization and potential to activate cytosolic nucleic acid sensing pathways following MOMP has never, to our knowledge, been investigated. “ – others have previously shown that BAX/BAK silencing prevents mitochondrial mt dsRNA release, implicating MOMP in this process (see PMID: 30046113), thus the authors should amend this statement.
 - We thank the reviewer for pointing out that Dhir et al. were the first to identify BAX- and BAK-dependent mtdsRNA release in the setting of PNPase knockout, which results in dramatically enhanced levels of mtdsRNA. We have updated our manuscript to reflect this fact, giving the authors appropriate credit for their findings (Lines 101-105), and edited Lines 41-47 to better reflect the question our study was focused on answering.

Reviewer #2 (Remarks to the Author):

In this manuscript, Killarney et al present a model where the pro-apoptotic executioner caspases inhibit mtRNA mediated activation of MAVS induced IRF3 and IFN β activation. They provide data to suggest this as a potential mechanism to target cancers where the cGAS/STING pathway is impaired by combining conventional chemotherapy with caspase inhibition. Altogether the data is strong and supports the conclusions of the paper. My main concerns are 1. whether the data strongly convinces that it is RNA released from mitochondria that triggers this pathway and 2. The lack of a mechanism of how caspase-3 and-7 inhibit the IRF3 pathway.

We thank the reviewer for their detailed and constructive feedback on our manuscript. In particular, we appreciated their questions regarding the specificity of mtRNA in activating the MAVS signaling pathway during CICD and their comments on the mechanistic link between casp3/7 and IFN signaling in our models. These points have been addressed with new analyses and experiments, and the text has been amended accordingly. In each of the below responses, we highlight new figures/figure panels and other content added during revision using red font. We believe these revisions have significantly improved the manuscript and thank the reviewer for his/her help in this process.

Major Points

1. To demonstrate that the activation of the IFN pathway in the absence of caspases is due to RNA release from the mitochondria, the authors a) deplete DNA with EtBr and b) block mtRNA production with IMT1B, an inhibitor of mitochondrial RNA polymerase. While the genes in the pathway of interest are not expressed under these conditions (Fig2f-h, Extended data Fig 2 c and f). However, these experiments lack controls for specificity. If DNA is depleted, then it would be expected that all expression would be impaired so measuring transcript levels does not address whether DNA is contributing to this pathway. Similarly, if RNA is depleted. The specificity of IMT1B for mitochondrial RNA should be demonstrated by including a positive control showing that nuclear transcription of genes not in this pathway is not impacted and stronger evidence is needed to show that mtDNA does not contribute in any way.
 - **We thank the reviewer for their comments on the specificity of ethidium bromide mtDNA depletion and IMT1B.**
 - i. **We want to point out to the reviewer that ethidium bromide treatment does not alter nuclear DNA levels, but rather it specifically depletes only mtDNA. The details of this technique and associated references are provided under the “Mitochondrial DNA Depletion” methods section (Lines 789-794). To clarify this distinction, we updated Fig. 2f to demonstrate that ethidium bromide pre-treatment does not decrease levels of the nuclear mRNA transcript *ISG15* at baseline. Additionally, Extended Fig. 2a shows no decrease in Hoechst nuclear DNA staining following ethidium bromide treatment. Finally, ethidium bromide-treated cells showed no difference in the nuclear housekeeping gene TBP compared to wild-type cells (data not shown).**
 - ii. **As was mentioned in our response to Reviewer 1, Major Point #2, we included a negative control experiment to establish that IMT1B does not lead to non-specific inhibition of IFN production upon a dsRNA challenge (Extended Fig. 2i) or to changes in mtDNA levels (Extended Fig. 2h).**
2. One of the main conclusions from the paper is that mtRNA and not mtDNA is the trigger for IRF3 signaling in the absence of caspases. However, apart from the experiments described in the above comment, there is not much evidence to support this. Ideally, an invitro experiment where the supernatant collected from isolated mitochondria treated with the drug cocktail (plus or minus IMTB1) is shown to trigger the IRF3 pathway in caspase-3/7 DKO extracts would answer this question but I am not certain if activation of this pathway can be measured in cell free extracts. This or a similar experiment should be considered and, at the very least, discussed.
 - **We thank the reviewer for their questions regarding mtRNA, not mtDNA, being the relevant ligand for MAVS pathway activation in STING-deficient cells undergoing CICD. We will summarize the data supporting this conclusion below.**
 - i. **mtDNA is known to activate Type I IFN signaling through two well-defined antiviral pathways (Riley and Tait, 2020 EMBO Reports): (1) the cGAS/STING/TBK1/IRF3 pathway, which recognizes cytosolic mtDNA, and (2) the TLR9/MyD88/IRF7 pathway, which recognizes endosomal mtDNA species. If mtDNA contributes to the IFN production, cGAS/STING or TLR9/MyD88 should be activated.**
 1. **The cellular models used for Figures 1-4 were deficient in the STING pathway, as evidenced by the lack of IFN production following treatment with cytosolic dsDNA agonists (Extended Fig. 2e-2f and Fig. 4a-4b). Additionally, we generated A375 STING^{-/-} cells. In all these models, we see robust IFN production in CICD, suggesting a STING-independent pathway.**
 2. **To rule out a role for the TLR9/MyD88 pathway in IFN production in this system, we included an experiment showing that IFN production was unchanged when MyD88 was knocked down (Extended Fig. 2g). Additionally, the pathway we identified is fully controlled by TBK1 and IRF3, which are not**

activated in the context of TLR9 (Kawasaki and Kawai, 2014 *Frontiers in Immunology*).

- ii. We identified the cytosolic dsRNA MDA5/MAVS/TBK1/IRF3 pathway as the driver of IFN production during CICD (Fig. 3). This finding implies that cells contain an endogenous dsRNA ligand for MAVS pathway activation. Known sources of endogenous dsRNA species include mtRNA, endogenous retroviruses, long interspersed nuclear elements, short interspersed nuclear elements, and dysregulated RNA splicing and metabolism products (Chen and Hur, 2022 *Nature Reviews Molecular Cellular Biology*).
 1. Our dataset supports mtRNA, and not other endogenous dsRNA species, as the likely dsRNA ligand for the following reasons:
 - a. PLX-4720 and S63845 treatment leads to the release of mtDNA species into the cytosol in a BAX- and BAK-dependent manner (Fig. 1c, Fig. 1d, Extended Fig. 1b-1c).
 - b. *BAX*^{-/-}*BAK1*^{-/-} knockout prevents IFN signaling during CICD, suggesting the relevant ligand is likely being released from the mitochondria rather than an indirect effect of our cytotoxic drug treatment (Fig. 2e).
 - c. The selective removal of mtDNA and, consequently, mtRNA from the cell prevents IFN signaling in CICD (Fig. 2f and Extended Fig. 2a-2d)
 - d. We validated that IMT1B is a specific inhibitor of mitochondrial RNA polymerase and does not alter mtDNA levels or IFN production upon extrinsic dsRNA agonism (Extended Fig. 2h-2i). We applied IMT1B to cells undergoing CICD and found that it decreases the phosphorylation of IRF3 and STAT1 (Fig. 2g and Fig. 3e), and significantly reduces *IFIT3* and *ISG15* mRNA production (Fig. 2h, Fig. 4d, and Fig. 5b)
 - Together, these data establish that mtRNA triggers IFN signaling in caspase-deficient conditions. That said, if the reviewer would like to suggest additional experiments that address concerns not addressed in the above, we would be happy to perform them.
3. The authors acknowledge that they do not know the targets of caspase-3 and-7 in this pathway but speculate that they could be MAVS and IRF3 based on their inclusion in the Degradase of caspase targets. Given that they have two very appropriate candidates, the authors should include an experiment to rule in or out these proteins as caspase substrates in this pathway.
 - We thank the reviewer for commenting on how caspase 3/7 negatively regulates Type I IFN production during CICD. We performed the recommended experiment (Extended Fig. 5a-5c). We found that apoptotic caspases cleave IRF3 and MAVS in A375 cells following treatment with PLX-4720 and S63845. Interestingly, the expression of both caspase-resistant IRF3^{D121/125A} and MAVS^{D429/490A} mutants in A375 cells did not lead to the activation of IFN signaling upon PLX-4720 and S63845 treatment. These data indicate that although MAVS and IRF3 are cleaved by caspases, they are insufficient to explain the suppression of mtRNA-IFN signaling in apoptosis.
 - i. We hypothesize that caspases degrade multiple members of the MAVS signaling pathway to robustly protect against IFN induction during apoptosis. We have plans to perform N-termini biotin labeling with subtiligase in apoptotic A375 wild-type and *CASP3*^{-/-}*7*^{-/-} cells to define all proteins degraded by caspases during cell death. To rescue mtRNA signaling in apoptosis, we hypothesize that caspase-resistant mutants of every protein involved in MAVS signaling must be expressed simultaneously in the

cell. While this is an ongoing topic of focus for our team, further investigation of caspase 3/7 targets is outside the scope of the current manuscript, a notion supported by the reviewers.

- ii. Additionally, it is worth noting that other manuscripts studying IFN signaling in CICD have not yet identified precisely how caspases suppress mitochondrial nucleic acid sensing in apoptosis. One potential explanation for this omission is that the relevant pathways are so thoroughly targeted by caspases 3/7 that extensive reconstitution by caspase-resistant mutants is required to rescue the effect. An alternative possibility is that inhibition of caspases slows the kinetics of cell death for long enough to allow IFN- β to be translated and released (a possibility raised initially in McArthur et al., 2018 Science). This alternative mechanism is in line with our observation that caspase-resistant IRF3 and MAVS mutants fail to rescue Type I IFN signaling in cell death. We have updated our discussion to illuminate these possibilities (Lines 322-334).

4. Figure 2c. The amount of IFN β released is 15pg/ml. This seems very small. The authors should include a positive control (a known inducer of IFN β) for comparison to show that this is a biologically relevant amount.
 - Thank you for this suggestion. In response, as a positive control, we transfected A375 cells with a dsRNA agonist (Extended Fig. 1g), yielding IFN- β production comparable to the amounts produced under CICD conditions. This is consistent with evidence that IFN- β production under CICD conditions is biologically relevant by inducing the ISG signature and an antiviral state. Additionally, the positive control dsRNA and dsDNA agonist experiments presented elsewhere in the manuscript yield ISG production that matches that observed in a CICD model (Fig. 2b and 2h, Extended Fig. 2f, and Fig. 4b). Also consistent with the biological relevance of IFN- β production during CICD is the fact that B16 mouse melanoma cells produce ~30pg/ml of IFN- β under the same conditions (Extended Fig. 4e). This concentration is sufficient to activate anti-tumor immunity *in vivo*. Finally, the 15 pg/ml IFN- β concentration in this assay, and the resultant downstream induction of ISGs, is consistent with other findings in similar models (White et al., 2014 Cell and Rongvaux et al., 2014 Cell).

Minor points

1. Casp3^{-/-} and casp7^{-/-} should be CASP as they are human genes
 - Thank you for catching this typographical error. We have amended the text throughout the manuscript to reflect this point.
2. Western blots (e.g Figure 1A) should show the full-length caspases and not just the cleaved fragment.
 - Thank you for this suggestion. We have included full-length caspases for all western blots in the manuscript (Fig. 1a).
3. The colors of the lines and symbols in Figure 5 d-g make it difficult to distinguish the different lines (especially in g where there are no symbols). The blue, dark blue and black lines look too similar, as do the orange, dark orange, and red.
 - Thank you for this suggestion. We have changed the colors to make them more legible (Fig. 5e-5i).

Reviewer #3 (Remarks to the Author):

A manuscript by Killarney et al. describes a series of experiments that aim to reveal the role of caspase 3/7 in the prevention of mt-dsRNA-driven Type I interferon response. The Authors confirmed and further explored previous observations that:

- (i) release of dsRNA from the mitochondrial matrix is BAX dependent (Dhir et al. Nature 2018)
- (ii) MDA5 is the primary sensing receptor of the mitochondrial dsRNA (Dhir et al. Nature 2018)
- (iii) mitochondrial nucleic acids are released by herniation of the inner mitochondrial membrane (McArthur et al., Science 2018)
- (iv) caspases can prevent mitochondria-stimulated Type I interferon (Rongvaux, A. et al. Cell 2014).

Still, the manuscript describes important advances in the field. This comprehensive and well-designed study summarizes the cascade of events from mt-dsRNA release to interferon β expression. Importantly, the Authors conclude that caspase 3/7 prevents activation of Type I interferon induction under mt-dsRNA release. Moreover, they show that inhibition of caspases combined with mt-dsRNA release from mitochondria can be used as an anti-cancer strategy. Overall, this is a well-written manuscript that describes technically sound experiments. Conclusions are well supported, and the manuscript is of general interest. I recommend its acceptance for publication in Nature Communications. I have one comment that Authors should modify their text to indicate that some of their observations have already been done by others. Otherwise, a reader of the manuscript can be misled. However, the manuscript still shows a high level of novelty.

- ***We want to thank the reviewer for their encouraging and supportive comments. In particular, we appreciate their succinct summary of the publications our manuscript is built upon before sharing how our findings are a significant and novel contribution to the field. We also agree that the findings of previous work should be clearly stated in our manuscript. To this end, we have re-read and modified the manuscript to ensure that the prior studies on which our work builds are clearly highlighted for readers. Specifically, we have made the following changes listed below in red font.***

Minor Points

1. Lanes 95-96: BAX-dependent mt-dsRNA release was described by Dhir et al. Nature 2018
 - **We have updated Lines 101-105 in accordance with this recommendation.**
2. Lanes 177-179: MDA5 as the primary pattern recognition receptor of mt-dsRNA was identified by Dhir et al. Nature 2018
 - **Lines 170-172 include the citation by Dhir et al. Nature 2018 regarding their identification of MDA5 as the primary PRR for mtdsRNA**
 - **We have updated Lines 198-199 to state that MDA5's control over mtdsRNA sensing in our model systems is consistent with that of the model described in Dhir et al. Nature 2018.**
3. Lanes 85-96: original idea of the role of the herniation of the inner mitochondrial membrane into the cytoplasm comes from McArthur et al., Science 2018
 - **Lines 36-38 credit the biology discovered by McArthur et al. Science 2018**
 - **Lines 101-105 reiterates the credit for their work.**
4. Lane 278-280: possible role of caspases in the regulation of mt-dsRNA triggered Type I interferon can be deduced from Rongvaux, A. et al. Cell 2014; this should be indicated in the text

- We amended **Lines 114-117** and **120-122** to state that our investigation of caspases' roles in IFN regulation during CICD is based on this previous work in mtDNA sensing.

Reviewer #4 (Remarks to the Author):

Killarney et al. reported that the leaked mtRNA during apoptosis activated type I IFN production. The authors found that chemotherapy-induced apoptosis induced IFN in tumors with defective cGAS/STING pathway. Using knockout cells, the authors found that mitochondrial double-stranded RNA activated IFN through MDA5, however, this finding is a recapitulation of previous work published in Nature (doi: 10.1038/s41586-018-0363-0) in 2018. They further found that mtRNA activation might be involved in anti-tumor activity using xenograft mouse models. Although the story integrates MOMP mtRNA leakage with mtRNA sensing and IFN activation, unfortunately, the novelty is moderate at the standing point of the reviewer.

We thank the reviewer for their insight into our manuscript. In particular, we appreciate the question regarding the role that mtDNA/cGAS/STING signaling may play in the context of our in vivo model system. This point has been experimentally addressed, and the manuscript has been updated accordingly, as described in further detail below. In each of the below responses, we highlight new figures/figure panels and other content added during revision using red font. Finally, while we agree that the findings published in 2018 by Dhir et al. laid the foundation from which our manuscript is built, and we have acknowledged their work to reflect this, we respectfully disagree that our work is a recapitulation of theirs or lacks novelty for the following reasons:

- 1. Dhir et al. investigated if mtDNA was a source of immunogenic dsRNA in vivo. They impressively identified that both genetic knockout and inactivating patient mutations in PNPase led to the initiation of MDA5/MAVS/IFN- β signaling dependent on cytosolic mtDNA leakage through BAX/BAK. This work proposes that PNPase plays a fundamental role in preventing mtDNA-dependent Type I Interferonopathy. Our current study expands upon the work of Dhir et al.. However, it focuses on a distinct research question: How does a cell avoid the detection of mtDNA in apoptosis following the chemotherapy-induced activation of BAX/BAK? Because our models, like most human tissues, contain functional PNPase and SUV3 proteins, it was conceivable at the outset that mtDNA levels would be low enough that MOMP would not lead to appreciable mtRNA release or IFN signaling. By demonstrating that MOMP indeed leads to an appreciable release of mtDNA that is sufficient, in caspase 3/7-depleted conditions, to activate Type I IFN signaling, our work reveals for the first time that a central function of caspase 3/7 is to suppress inflammatory signaling secondary to mtRNA release during MOMP. This has never, to our knowledge, been shown.***
- 2. After discovering that caspases-3 and -7 suppress mtDNA-dependent MAVS signaling during apoptosis, we identified a translationally important context for leveraging this described signaling pathway. Namely, we first revealed that widely used targeted and cytotoxic chemotherapies could activate immunotherapeutic Type I IFN signaling in the setting of caspase 3/7 inhibition. Second, building on recent literature identifying a large subset of immunotherapy-resistant cancers marked by their repression of cGAS or STING, we highlighted the concept that the pathway identified in our studies can activate IFN signaling even in tumor cells lacking cGAS/STING competence. Together, these data imply that pairing diverse, FDA-approved, apoptosis-inducing cancer therapies with a caspase inhibitor like Emricisan – which is well tolerated in humans – may be a broadly useful strategy for inducing tumor-expressed IFN- β signaling that has near-term translational potential. This, too, has never been shown.***
- 3. Our work is the first to demonstrate that mtDNA can contribute to CD8+ T cell-dependent anti-tumor immunity in vivo. Furthermore, we found that this phenotype can restore anti-PD1 immunotherapy***

responsiveness in an immunologically cold melanoma tumor model. Given the points raised in (2) above, this novel finding may have significant near-term clinical relevance.

Major Points:

1. It seems that only three mtRNA were examined by RT-qPCR for the mtRNA leakage assays. There are >30 mtRNA genes. What is the rationale to cherry pick these two gene?
 - **We thank the reviewer for their comment regarding our decision-making on our choice of the mtRNA genes to assay. We chose these three genes to represent global mtRNA transcription to align with previous reports (*Dhir et al. Nature 2018* and *Bonekamp et al. Nature 2020*). In response to this comment, we added three genes to this panel, with similar findings (**Extended Fig. 1b**).**
2. In fig2b, the ISG15 mRNA level in control cells is very low; however, there is about 60-fold of ISG15 in the control cells in fig 3b.
 - **We want to point out to the reviewer that in Fig. 2b, control cells are A375 wild-type cells. In Fig. 3b, control cells are A375 CASP3^{-/-}7^{-/-} cells which were properly controlled for the additional CRISPR/Cas9 knockout. Both descriptions are found within the figure legends.**
3. According to the data, wild type cells had a low IFN response due to the caspases. But in reality, most tumors have intact caspases. How do the author interpret it immunotherapy?
 - **We thank the reviewer for their question regarding the translatability of our finding. We agree that most tumors have functional caspases and do not activate mtRNA-dependent IFN signaling upon apoptotic stimuli. Our manuscript argues that our findings provide a rationale for pairing apoptosis-inducing targeted and cytotoxic chemotherapies with a caspase inhibitor, such as Emricasan, as a synergistic anti-cancer strategy. We hypothesize that future studies pairing Emricasan with targeted and cytotoxic chemotherapies will yield caspase-independent cell death to both directly kill cancer cells and activate mtRNA-dependent anti-tumor immunity.**
4. What is mechanism of how caspase3/7 inhibits IFN? There are several papers that have shown that caspases can inhibits IRF3 signaling. Previous work should be discussed.
 - **Please refer to Reviewer 2, Main Point #3 for a detailed discussion regarding the mechanism of caspase 3/7's inhibition of IFN signaling. We confirmed that MAVS and IRF3 are cleaved by caspases in our apoptotic models, but that does not fully explain the suppression of mtRNA signaling during apoptosis (**Extended Fig. 5a-5c**). Our discussion acknowledges previous publications identifying how caspases regulate MAVS and IRF3 and has been updated to reflect our recent findings (**Lines 314-334**).**
5. B16 cells have the functional cGAS/STING pathway. The xenograft experiments cannot exclude the effects of mtDNA, which might play a major role.
 - **We thank the reviewer for their insightful question regarding the relative contribution of mtDNA and mtRNA to our *in vivo* anti-tumor immunity mechanism. This question recognizes that while the studies presented in Figures 1-4 of our manuscript use model systems that are cGAS/STING deficient and thus allow for a clean assessment of mtRNA-driven MDA5/MAVS/IFN- β signaling, the studies in Figure 5 utilize a model system (B16) that is both MDA5/MAVS- and cGAS/STING-competent. In response, we have updated both our *in vitro* and *in vivo* studies with the B16 model as follows:**
 - i. ***In Vitro***
 1. **Using EtBr pre-treatment, we confirmed that the entirety of the *in vitro* IFN signaling seen in B16 cells under CICD conditions is dependent on tumor-intrinsic**

mitochondrial nucleic acids (Fig. 5a). IMT1B treatment reduced ISG production in B16 cells undergoing CICD, validating a significant contribution to this IFN signaling from mtRNA (Fig. 5b). Finally, we found that STING and MAVS knockout yielded equal reduction of both phosphorylation of TBK1 and *IFIT3* mRNA reduction in CICD-induced B16 cells (Extended Fig. 4j and Fig. 5c). Taken together, these data suggest that mtDNA and mtRNA contribute approximately equally to Type I IFN production in CICD-induced B16 cells.

ii. *In Vivo*

1. We previously established that the *in vivo* phenotype entirely depends upon IFN signaling, based on our work identifying tumor-intrinsic IRF3 as necessary for the anti-tumor response (Fig. 5f).
2. We demonstrated in our revised manuscript that tumor-intrinsic STING and MAVS contribute to the Type I IFN-dependent anti-tumor response, suggesting that both mtDNA and mtRNA contribute to the *in vivo* mechanism (Fig. 5g).

In sum, our studies demonstrate that both the mtDNA-dependent cGAS/STING and mtRNA-dependent MDA5/MAVS pathways can contribute to tumor-intrinsic Type I IFN production under CICD conditions when both pathways are intact. We have updated our manuscript to reflect this fact (Lines 255-286 and Fig 5 Heading).

6. Many papers about mtRNA are not discussed.

- We thank the reviewer for their suggestion regarding citations of mtRNA literature. We want to ensure that all relevant mtRNA papers are correctly cited, and we do not intend to exclude any such papers. To this end, we have highlighted those papers that we believe most directly provide the foundation for our work. However, we welcome any additional suggestions the reviewer may have, as we want to ensure that prior work in the field is appropriately credited and highlighted.

Reviewers' Comments:

Reviewer #1:

Remarks to the Author:

The authors have comprehensively addressed all my comments.

Reviewer #2:

Remarks to the Author:

In the revised manuscript by Killarney et al, the authors have successfully addressed all my comments. This is an important addition to the growing body of evidence on the pro-inflammatory effects of inducing CICD.

Reviewer #3:

Remarks to the Author:

The authors have adequately addressed all the issues raised. I recommend the publication of their interesting work.

Reviewer #4:

Remarks to the Author:

The authors satisfactorily address my concerns.

Point-by-point responses to reviewers

We thank the reviewers for their positive appraisals of our work and their detailed and constructive questions.

Reviewer #1 (Remarks to the Author):

The authors have comprehensively addressed all my comments.

We thank the reviewer for their helpful comments, which led to significant improvements in our manuscript.

Reviewer #2 (Remarks to the Author):

In the revised manuscript by Killarney et al, the authors have successfully addressed all my comments. This is an important addition to the growing body of evidence on the pro-inflammatory effects of inducing CICD.

We appreciate the compliments of our manuscript and thank the author for their insightful feedback on our findings.

Reviewer #3 (Remarks to the Author):

The authors have adequately addressed all the issues raised. I recommend the publication of their interesting work.

We want to thank the reviewer for the positive and encouraging comments they made on our manuscript.

Reviewer #4 (Remarks to the Author):

The authors satisfactorily address my concerns.

We would like to thank the reviewer for their comments and appreciate how they enhanced the conclusions of our manuscript.